

# Effect of salt stress on different tiller positions in rice and the regulatory effect of prohexadione calcium

Rongjun Zhang[1], Dianfeng Zheng[1,2,3], Naijie Feng[1,2,3], Linfeng Linfeng[1], Jinning Ma[1], Xiayi Yuan[1], Junyu Huang[1] and Lisha Huang[1]

[1] College of Coastal Agricultural Sciences, Guangdong Ocean University, Zhanjiang, Guangdong, China
[2] South China Center of National Saline-Tolerant Rice Technology Innovation Center, Zhanjiang, Guangdong, China
[3] Shenzhen Research Institute of Guangdong Ocean University, Shenzhen, Guangdong, China

Corresponding authors
Dianfeng Zheng,
zhengdf@gdou.edu.cn
Naijie Feng, fengnj@gdou.edu.cn

## ABSTRACT

Soil salinization has resulted in a significant decrease in crop yields, particularly affecting the production of crops like rice (*Oryza sativa* L.). Prohexadione calcium (Pro-Ca) can enhance crop resilience against failure by managing plant height. However, its impact on various tiller positions during the tillering phase of rice under salt stress remains unknown. This study explores the distinct effects of salt stress on the physiological traits of the main stem and different tiller segments of rice plants, along with the role of Pro-Ca in mitigating salt stress. The findings revealed that under salt stress conditions, the number of tillers and leaves on the main stem decreased significantly in rice. Moreover, the levels of malondialdehyde (MDA) and $H_2O_2$ in the leaves and stems of each tiller position notably increased. The percentage of tillers experiencing reduction or elevation was higher than that of the main stem compared to the respective control. Application of Pro-Ca through foliar spraying under NaCl stress effectively alleviated the impact of salt stress on the tiller growth of rice during the tillering phase. It also boosted the activities of antioxidant enzymes like superoxide dismutase (SOD) and peroxidase (POD) in the leaves and stems of the tillers. Furthermore, it successfully mitigated the damage inflicted by salt stress on the cell membrane of rice tillers during the tillering phase. The regulatory effect of calcium on cyclic acid was particularly pronounced in alleviating the impact on the tillers under salt stress conditions.

## INTRODUCTION

In recent years, soil salinization has become a global issue due to rising sea levels and expanding salinized land areas caused by global warming (*Ahmed et al., 2021*), severely reducing global crop yields and agricultural production (*Munns & Tester, 2008*; *Dai et al., 2022*). Enhancing crop salt tolerance and effectively addressing the damage from salt stress on yield has emerged as a critical research focus (*Wang et al., 2024*). Salt stress is a key abiotic factor that impacts the growth of most plants. In general, salt concentrations lead to

changes in physiological and biochemical functions, limiting the growth and development of above-ground plant parts and root systems (*dos Santos et al., 2022*), when soil salinity increases, the water potential of the soil solution decreases below that of the plant root cells. This makes root uptake challenging and results in osmotic stress. Osmotic stress leads to the closure of plant stomata, hindering $CO_2$ uptake, weakening photosynthesis, causing nutrient deficiencies. The accumulation of $Na^+$ and $Cl^-$ in cells affects mineral uptake and transport, inhibits enzyme activity, and leads to dehydration of plant cells (*Huang et al., 2023*). Salt stress can also result in the accumulation of reactive oxygen species (ROS), damaging cellular structures and biomolecules, thus limiting the growth of major crops like rice (*Hussain et al., 2018*). Prolonged exposure to harsh environments has led to the evolution of a range of salt tolerance mechanisms in crops, including: changes in morphology, water relations, photosynthesis, hormones, ion distribution and biochemical adaptations (*dos Santos et al., 2022*), within certain limits, allow them to competitively obtain water from the soil and maintain nutrient balances in the body in response to ionic stresses thereby surviving adverse soil conditions (*Hoang et al., 2016*).

Rice (*Oryza sativa* L.) is a moderately salt-sensitive crop (*Joseph, Jini & Sujatha, 2010*), and its growth and development are severely affected by salt stress (*Zhang et al., 2012*), and this effect varies depending on the developmental stage, the degree and duration of stress, and the variety (*Zheng et al., 2023*). It has shown that the effects of salt stress on rice germination and emergence are mainly characterized by a reduction in germination rate, germination speed, and germination energy, leading to a reduction in shoot length, root length, and dry weight of rice (*Taratima, Chomarsa & Maneerattanarungroj, 2022*). Salt stress during the seedling period is mainly manifested in the damage to leaves and root system (*Chang et al., 2019*). Salt stress affects rice tillering mainly by reducing tillering capacity and delaying the reproductive process, and the duration of delay is positively correlated with the degree of salt stress, and primary and secondary tillers being more affected than the main stem. Additionally, salt stress reduces soil fertility and causes nutrient imbalance, and salinity stress inhibits nutrient uptake by the root system, ultimately leading to reduced tillering or tiller death due to nutrient deficits (*Ruan, Hu & Schmidhalter, 2008*). Primary stems and primary tillers contribute more to crop yield than secondary tillers due to asymmetric competitive advantages under stress conditions, and these advantages are associated with increased leaf number. Transportation of water and nutrients between the primary stem and tiller through the vascular bundles at the tiller nodes is essential for tiller development and survival (*Yang et al., 2022*). Salt stress in the formation of young spikes and spiking and flowering stage of rice is mainly manifested in the following ways: yellowing of leaves, delayed spiking, prolonged spiking period, increase in the number of degradation of glumes, shorter spike lengths, decrease in the number of solid grains, less full grains, more black rotting of roots in the late stage, early senescence, and ultimately affecting the yield of rice (*Chang et al., 2019*).

Plant growth regulators, as organic compounds with effects on growth and development similar to natural plant hormones, control plant growth by initiating various physiological and metabolic processes (*Kaya et al., 2023*; *Zhao et al., 2023*). The formation of the endogenous plant hormone gibberellin requires hydroxylase enzymes to catalyze a series of

hydroxylation reactions, and these hydroxylases require 2-ketoglutarate as a coenzyme. Prohexadione calcium (Pro-Ca) imitates the coenzymes' structure and competitively hinders their function, thus impeding the synthesis of active gibberellins. Among these hydroxylation reactions, the reaction pathway for the formation of GA1 is the most sensitive to Pro-Ca, whereas the pathway for the formation of GA4 is not involved in the β-hydroxylation reaction, so that Pro-Ca selectively inhibits the synthesis of gibberellin GA1. GA1 is mainly found in the nutrient organs, controlling the elongation and growth of stems and leaves, while GA4 is mainly found in the reproductive organs, controlling flower bud differentiation and hot grain development. Pro-Ca is an ideal dwarfing agent because of its strong synthetic activity in inhibiting GA1. Pro-Ca inhibits active gibberellin synthesis while protecting the activity of both surviving gibberellins, so Pro-Ca has dual activity on gibberellin metabolism (*Kim et al., 2010*; *Ilias & Rajapakse, 2005*). Pro-Ca has been shown by previous authors to have specific regulatory effects on rice, apple, strawberry, *etc.*, (*Kim et al., 2010*, *2007*; *Lee, Foster & Morgan, 1998*). In previous studies, researchers have explored the risks of salt stress, the characteristics of tillering, and the mechanism of action of Pro-Ca acid. Our own research has shown that Pro-Ca acid can mitigate the harm caused by NaCl to the antioxidant capacity, photosynthetic properties, and cell membranes during the tillering stage in rice (*Zhang et al., 2023a*; *Huang et al., 2023*). However, further investigations are necessary to understand the varying impacts of salt stress on the main stem and tiller, as well as the regulatory function of Pro-Ca.

In this study, we aimed to investigate the differential effects of salt stress on rice main stems and tillers and the regulatory role of calcium switched acid by comparing the relevant morphology building indexes, antioxidant enzyme activities, membrane damage indexes, and soluble protein contents in leaves and stems of rice main stems, first tillers and second tillers at the tillering stage under different treatments.

## MATERIALS AND METHODS

### Materials and reagents

Huanghuazhan (conventional rice) was provided by Longping Seed Co. Ltd (Hunan, China), and Xiangliangyou900 (hybrid rice) was provided by Nianfeng Seed Science and Technology Co. Ltd. (Hunan, China).

The original solution of the test regulator 5% Pro-Ca used in this experiment was provided by Sichuan Runer Technology Co. Ltd. (Chengdu, Sichuan).

### Experimental designs

Full and uniform rice seeds were selected, sterilized with 3% $H_2O_2$ for 15 min and then washed repeatedly with distilled water, distilled water was added until the seeds were submerged, and the seeds were soaked for 24 h at 30 °C, after which they were germinated under dark conditions for 24 h. The experiment was selected to be carried out in the daylight linkage greenhouse of the College of Coastal Agriculture, Guangdong Ocean University, and the germinated seeds were uniformly sown on the rice-planting trays (specifications of 28–30 cm × 58–60 cm), about 5–8 seeds in each hole, and the soil used for seedling was a 3:1 mixture of brick red soil and nutrient soil.

After transplanting cultivation using caliber × bottom diameter × height of 19 × 15 × 18 cm plastic pots, each pot containing 3 kg of sun-dried soil, before transplanting a fixed amount of each pot to add 1 L of water, to be stabilized when the water surface line marking, and regularly replenish water to maintain the water layer. When the seedlings in the seedling tray were three leaves and one heart, the seedlings with consistent growth were selected and transplanted, and the depth of transplanting was about 1.5 cm, with three holes in each bucket and one plant in each hole. After the end of greening and before tillering, select the evening of sunny weather at about 16:00 to carry out regulator treatment through foliar spraying, about 10 ml per pot, to ensure that the front and back of the leaf spraying evenly, in order to ensure its normal absorption. The regulator treatment was followed by a 0.3% salt treatment 48 h later. Tagging and tracking marking of tiller occurrence. Tillers were labeled with secondary and leaf positions. Ensure spatial distance between seedlings at each sampling to prevent competition between individuals due to different spatial size.

The experiment was set up with eight treatments, Xiangliangyou900 variety included four treatments as follows: XCK (distilled water), XS (0.3% NaCl), XPro-Ca (100 mg·L$^{-1}$ Pro-Ca), XPro-Ca+S (100 mg·L$^{-1}$ Pro-Ca + 0.3% NaCl), and Huanghuazhan variety included four treatments as follows: ZCK (distilled water), ZS (0.3% NaCl), ZPro-Ca (100 mg·L$^{-1}$ Pro-Ca), and ZPro-Ca+S (100 mg·L$^{-1}$ Pro-Ca + 0.3% NaCl), with three replications per treatment. The leaves and stems of the main stem, the first tiller, and the second tiller were taken every 7 d (7, 14, 21, 28, and 35 d after salt treatment) for the determination of related indexes.

## Determination of morphological indices

Morphological indexes such as plant height, root length, and number of tillers were measured directly by using vernier calipers to measure the intersection of stem and root to determine the stem base width, and by using a leaf area meter (YX-1241) to measure the inverted two leaves and inverted three leaves of each tiller position, and by using the conventional drying method, the samples of each part of each treatment were killed for several and a half times for 30 min in an oven at 105 °C, and then dried at 80 °C to a constant weight and then determined the dry weight.

## Determination of antioxidative enzyme activities

At 7, 14, 21, 28, and 35 d after NaCl stress, the leaves and stems of 15 rice plants with different tiller positions were rapidly frozen in liquid nitrogen, and then stored in −80 °C, respectively. Three replicates of 0.5 g samples (leaves and stems weighed separately) were ground in liquid nitrogen, and then 10 ml of pre-cooled phosphate buffer (0.05 mM PBS, pH 7.8) was added, ground to homogenate, and then centrifuged at 6,000 × g for 20 min at 4 °C, and the supernatant was aspirated and set aside in a 4 °C refrigerator.

Mixing 3 ml of reaction solution (50 ml PBS pH 6.0 + 28 μL guaiacol + 19 μL H$_2$O$_2$ at 30% concentration) with 40 μL of supernatant. The absorbance was recorded every 30 s for four times and the dynamic absorbance was measured at 470 nm using a spectrophotometer (GENESYS 180 UV-Vis; Thermo Fisher Scientifc, Waltham, MA,

USA) to determine the peroxidase (POD, EC 1.11.1.7) activity, 1 unit of enzyme activity for each minute of OD increase of 0.01.

Taking 0.1 ml of supernatant and mix with 2.9 ml of reaction solution (100 ml of PBS pH 7.0 + 0.05 ml of $H_2O_2$ at 30% concentration) and the absorbance at 240 nm was measured using a spectrophotometer (GENESYS 180 UV-Vis; Thermo Fisher Scientific, Waltham, MA, USA) and recorded every 30 s for four times. CAT (EC 1.11.1.6) activity was calculated using 0.01 decrease in OD per minute as a unit of enzyme activity (*Aebi, 1984*).

Superoxide dismutase (SOD, EC 1.15.1.1) activity was determined using the nitro blue tetrazolium (NBT) method (*Giannopolitis & Ries, 1977*). Adding 0.1 ml of supernatant to 2.9 ml of reaction mixture (2.61 ml MET + 0.097 ml EDTA-Na$_2$ + 0.097 ml NBT + 0.097 ml riboflavin) and irradiated for 20 min at 4,000 lux light at 25 °C. At the end of the reaction the absorbance at 560 nm was measured using the solvent in the unilluminated cuvette as a control tube, and the total activity of SOD was also calculated using 50% inhibition of NBT photochemical reduction as one enzyme activity unit (U).

Mixing 0.1 ml of supernatant with the reaction solution (2.6 ml EDTA-Na$_2$ + 0.15 ml AsA + 0.15 ml $H_2O_2$) and the absorbance at 290 nm was determined by spectrophotometer recording every 30 s for 4 times. APX (EC 1.11.1.11) activity was calculated in absorbance of 0.01 in 1 min is defined as one unit of enzyme activity u, and enzyme activity is expressed as u/g (FW) (*Nakano & Asada, 1981*).

## Determination of membrane damage index

Malondialdehyde (MDA) content was determined by TBA method (*Guo et al., 2018*). Adding 10 ml of phosphate buffer (0.05 mM PBS, pH 7.8) to 0.5 g of sample and ground, then centrifuged at $10,000\times g$ for 10 min at 4 °C. A total of 1 ml of the supernatant was taken and mixed with 2 ml of 0.6% TBA (thiobarbituric acid) in a centrifuge tube. The mixture was boiled in a boiling water bath for 15 min and then centrifuged at $10,000\times g$ and 25 °C for 10 min. The absorbance of the supernatant was measured spectrophotometrically at 450, 532, and 600 nm, respectively. The MDA content was calculated according to equation: MDA content (mmol·$g^{-1}$ FW) = [6.452 × ($OD_{532}$ − $OD_{600}$) − 0.559 × $OD_{450}$] × Vt/(Vs × W), where Vt is the total volume of extract, Vs is the volume of extract used for the assay, and W represents the fresh weight of the sample.

A total of 0.5 g of the sample was taken, 5 ml of 0.1% TCA solution was added, ground in liquid nitrogen and centrifuged at $10,000\times g$ for 10 min. Then, 0.5 ml of the supernatant was added to 0.5 ml of 10 mM PBS buffer and 1 ml of KI solution, and the reaction was carried out in the dark at 28 °C for 1 h. The $H_2O_2$ content of the sample was determined by spectrophotometric (GENESYS 180 UV-Vis; Thermo Fisher Scientific, Waltham, MA, USA) at 390 nm to determine the $H_2O_2$ content by measuring the absorbance (*Jessup, Dean & Gebicki, 1994*).

Subcellular localisation staining of plant leaf tissues for $O_2^-$ and $H_2O_2$ was carried out with a slight modification of the method of *Romero-Puertas et al. (2004)*.

For histochemical detection of analytical $O_2^-$: fresh leaves were washed with distilled water, evacuated, and placed in 10 mL centrifuge tubes containing 0.1% (w/v) nitrogen

blue tetrazolium (NBT, pH 7.8) for 24 h. Leaves were rinsed with distilled water, 95% ethanol was added, and then boiled in a boiling water bath for 40 min, destained, and photographed.

Histochemical detection of $H_2O_2$: Fresh leaves were washed with distilled water, evacuated, and soaked in 1 mg/L diaminobenzidine 13,3′-tetrahydrochloride (DAB, pH 3.8) in a 10 mL plastic centrifuge tube for 24 h. Leaves were rinsed with distilled water, then boiled in 95% (v/v) ethanol for 40 min, decoloured, and photographed.

Histochemical staining of leaf cell death was referred to *Schraudner et al. (1998)*. The treated leaves were washed with distilled water, and the leaves were immersed in 0.25% (w/v) Evans blue solution for 24 h. The leaves were removed, rinsed with distilled water and placed in boiling anhydrous ethanol:glycerol (9:1) for 30 min, and chlorophyll was removed until the underside of the leaf appeared white and photographed.

## Determination of the soluble protein content

To determine the soluble protein content, the method of *Bradford (1976)* was used with the Caumas Brilliant Blue G-250 staining method, 0.5 g of the sample was added to 10 ml of 0.05 mol/L pre-cooled phosphate buffer (pH 7.8) and ground in liquid nitrogen, and centrifuged at $12,000 \times g$ at 4 °C for 20 min, and the supernatant was the crude protein extract. The protein content was determined by adding 1 ml of enzyme solution to 5 ml of koammas brilliant blue solution and then shaking well, and the absorbance value at 595 nm was measured after 2 min of reaction.

## Statistical analyses

Using Excel 2016 for statistical analysis and SPSS 25.0 (IBM, Armonk, NY, USA) for further examination, one-way ANOVA and Duncan's method were employed for conducting ANOVA and multiple comparisons. The outcomes were presented as mean (X) ± standard error (SE). Graphs were created using Origin 2018 software, with distinct lowercase letters denoting significant differences between treatments ($p < 0.05$).

## RESULTS

### Effect of salt stress on morphological indexes at the tillering stage of rice and regulation by Pro-Ca

From the experimental results, it was found that salt stress negatively affected the tillering ability of both rice varieties (Figs. 1A and 1B), and the number of tillers decreased by 30.0–44.43% and 12.52–33.35% in Xiangliangyou900 and Huanghuazhan, respectively, from the 7th to the 35th days (Figs. 1C and 1D). The number of main stem leaves of Xiangliangyou900 decreased by 9.09%, 5.72%, 15.00%, and 2.04% on the 14th, 21st, 28th, and 35th days, respectively, and that of Huanghuazhan decreased by 6.90%, 21.05%, 11.11%, 12.20%, and 8.16% on the 7th, 14th, 21st, 28th, and 35th days, respectively, after NaCl stress (Figs. 1E and 1F).

The plant heights of Xiangliangyou900 and Huanghuazhan were reduced by 4.07–11.60% and 10.31–28.10%, respectively, compared with the control from the 7th to the 35th days after salt stress (Table 1). The first tiller length of both varieties was reduced

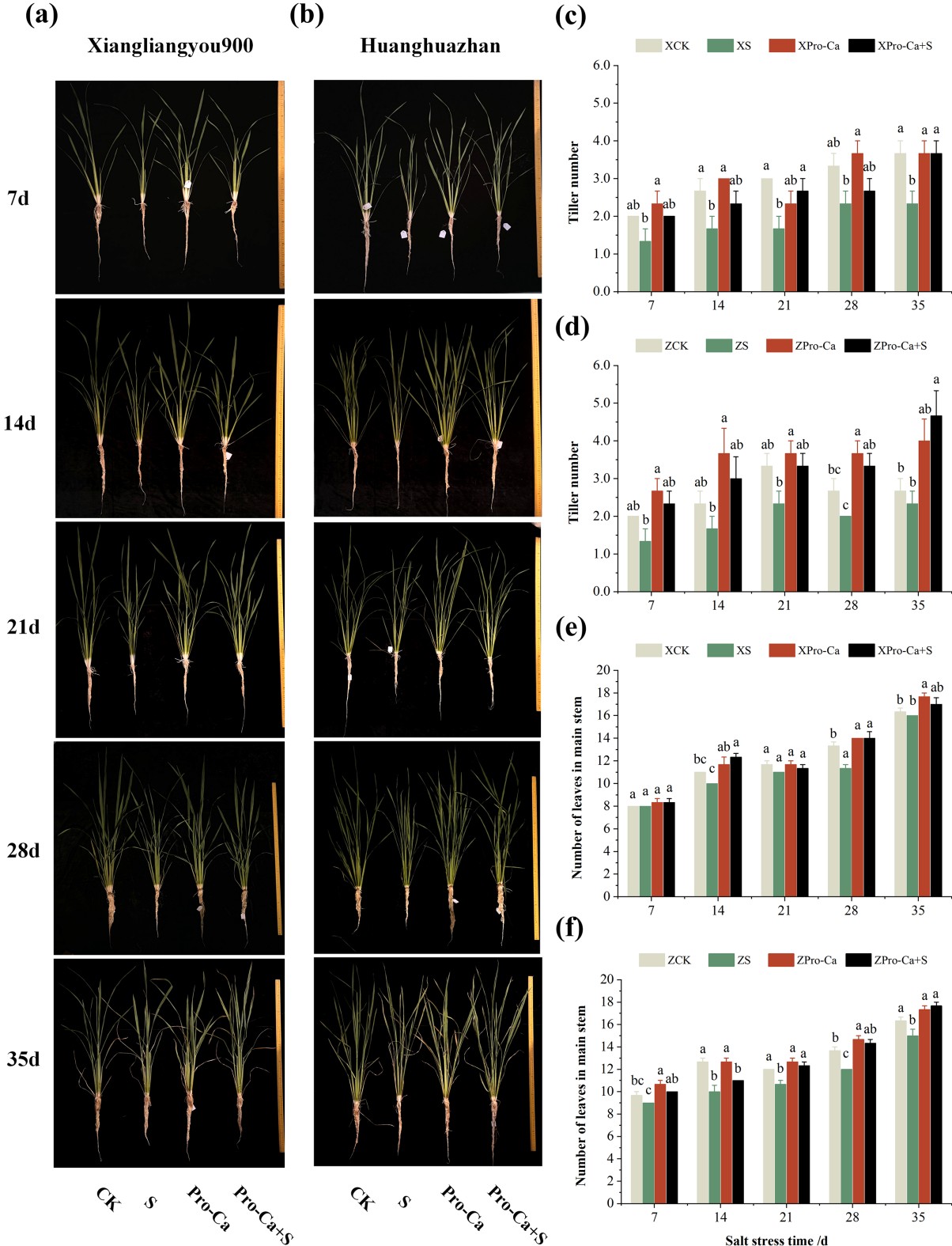

**Figure 1 Effects of Pro-Ca on rice growth under salt stress.** (A, B) Plant growth. Rice morphology after 7, 14, 21, 28, and 35 d of salt stress. (A) Shows the form of Xiangliangyou900, and (B) shows Huanghuazhan. From left to right, the plants were treated as follows: CK (distilled water), S (0.3% NaCl), Pro-Ca (100 mg·L$^{-1}$ Pro-Ca), Pro-Ca+S (100 mg·L$^{-1}$ Pro-Ca + 0.3% NaCl). (C, D) Tiller numbers of Xiangliangyou900 and

**Figure 1 (continued)**
Huanghuazhan in the main stem between different treatments. (E, F) Leaf numbers in the main stem. Comparison of tiller numbers and leaf numbers in the main stem between different treatments. Values are means ± SD ($n = 3$) and bars indicate SD. Columns with different letters indicate significant difference at $P < 0.05$ (Duncan's test). Xiangliangyou900: XCK (distilled water), XS (0.3% NaCl), XPro-Ca (100 mg·L$^{-1}$ Pro-Ca), XPro-Ca +S (100 mg·L$^{-1}$ Pro-Ca + 0.3% NaCl), Huanghuazhan: ZCK (distilled water), ZS (0.3% NaCl), ZPro-Ca (100 mg·L$^{-1}$ Pro-Ca), ZPro-Ca+S (100 mg·L$^{-1}$ Pro-Ca + 0.3% NaCl).

by 2.34% to 29.22% and 13.02% to 38.46%, respectively, compared with the CK from the 7th to the 35th days after salt stress (Table 1). Compared with the main stem and the first tiller, the effect of NaCl stress on the length of the second tiller was more significant, and the length of the second tiller of Xiangliangyou900 and Huanghuazhan was reduced by 12.70–31.86% and 19.60–39.38% from the 7th to the 35th d after salt stress, respectively (Table 1). The stem base width of the main stem of Xiangliangyou900 was reduced by 16.45% to 25.68% from 7th to 35th days after salt stress compared with the control, and that of Huanghuazhan was reduced by 17.41% to 38.99% from 7th to the 35th days after NaCl stress (Table 1). Salt stress also significantly reduced the stem base width of the first and second tillers of both varieties. Compared with the CK treatment, the stem base width of the first tiller was reduced by 16.42–44.02% and 7.74–43.86%, and the stem base width of the second tiller was reduced by 13.63–36.32% and 26.62%–43.46%, respectively, in Xiangliangyou900 and Huanghuazhan from the 7th to the 35th days after the salt stress (Table 1).

The main stem leaf area of Xiangliangyou900 was reduced by 19.83% to 41.70% from the 7th to the 35th days after salt stress compared with the control, and the main stem leaf area of Huanghuazhan was reduced by 29.74% to 45.76% from the 7th to the 35th days (Table 2). Compared with the control, the leaf area of the first tiller decreased by 36.83% to 60.23% and 30.66% to 73.22%, and the leaf area of the second tiller decreased by 30.63% to 43.52% and 43.26% to 65.35%, respectively, in Xiangliangyou900 and Huanghuazhan from the 7th to the 35th days after salt stress (Table 2). In addition, compared with the CK, the root length of Xiangliangyou900 decreased by 25.73% to 48.08% from the 7th to the 35th days after salt stress, and the root length of Huanghuazhan decreased by 15.75% to 34% from the 7th to the 35th days after NaCl stress, which were significant differences (Table 2).

As shown in Table 3, the aboveground dry weight of the main stems of Xiangliangyou900 and Huanghuazhan decreased by 27.53% to 62.62% and 29.77% to 52.86%, respectively, compared with the control from the 7th to the 35th d after NaCl stress. Compared with CK, the dry weight of the first tiller decreased by 9.86% to 63.22% and 30.39% to 65.68%, and the dry weight of the second tiller decreased by 8.44% to 37.46% and 20.99% to 65.03%, respectively, in Xiangliangyou900 and Huanghuazhan from the 7th to the 35th days after salt stress (Table 3). In addition, the root dry weight of Xiangliangyou900 decreased by 34.32% to 70.88% from the 7th to the 35th days after NaCl stress compared with the control, and which of Huanghuazhan decreased by 36.33% to 70.21% (Table 3).

We can see from Fig. 1 that exogenous foliar application of Pro-Ca effectively alleviated the inhibitory effect of NaCl stress on the growth parameters of the two varieties. Foliar

**Table 1 Effects of Pro-Ca on plant height and stem base width of rice main stem, first tiller, and second tiller stems at the tillering stage under salt stress.**

| Time/d | Treatment | Plant height/cm | | | Stem base width/mm | | |
|---|---|---|---|---|---|---|---|
| | | 0 | I | II | 0 | I | II |
| 7 | XCK | 56.4 ± 0.4a | 41.3 ± 0.9a | 36.5 ± 1.2a | 9.4 ± 0.6b | 8.0 ± 0.2a | 5.2 ± 0.1b |
| | XS | 52.4 ± 0.9b | 29.2 ± 2.7b | 27.3 ± 1.9c | 7.1 ± 0.2c | 5.6 ± 0.4b | 4.5 ± 0.3b |
| | XPro-Ca | 48.7 ± 0.9c | 28.9 ± 0.4b | 31.7 ± 0.7b | 11.2 ± 0.5a | 8.3 ± 0.2a | 6.2 ± 0.2a |
| | XPro-Ca+S | 51.2 ± 0.4b | 29.9 ± 2.0b | 35.0 ± 1.0ab | 9.4 ± 0.3b | 7.3 ± 0.6a | 6.2 ± 0.3a |
| | ZCK | 63.2 ± 2.7a | 42.9 ± 2.7a | 45.2 ± 5.0a | 9.2 ± 0.4a | 7.9 ± 0.5b | 6.4 ± 0.1a |
| | ZS | 45.5 ± 1.6c | 26.4 ± 1.7c | 27.4 ± 1.9b | 6.3 ± 0.2b | 5.2 ± 0.2c | 3.6 ± 0.3b |
| | ZPro-Ca | 51.6 ± 0.6b | 34.1 ± 2.3b | 31.6 ± 2.3b | 9.7 ± 1.2a | 8.9 ± 0.1a | 6.3 ± 0.1a |
| | ZPro-Ca+S | 56.5 ± 1.3b | 36.5 ± 1.2ab | 35.2 ± 2.0ab | 8.5 ± 0.1a | 7.2 ± 0.2b | 6.0 ± 0.4a |
| 14 | XCK | 66.5 ± 2.1a | 46.9 ± 1.0a | 43.6 ± 1.0a | 10.8 ± 0.7a | 6.9 ± 0.4a | 6.7 ± 0.3a |
| | XS | 58.2 ± 0.2b | 45.8 ± 1.6a | 29.7 ± 1.5c | 8.1 ± 0.2b | 5.8 ± 0.1b | 4.6 ± 0.5b |
| | XPro-Ca | 56.5 ± 0.7b | 41.7 ± 1.0b | 33.7 ± 3.9bc | 12.4 ± 0.5a | 7.3 ± 0.4a | 6.7 ± 0.1a |
| | XPro-Ca+S | 52.3 ± 0.3c | 32.6 ± 0.9c | 37.4 ± 0.6ab | 11.7 ± 0.9a | 7.7 ± 0.1a | 6.8 ± 0.1a |
| | ZCK | 67.3 ± 1.9a | 56.3 ± 2.0a | 51.8 ± 2.6a | 9.9 ± 0.4a | 8.0 ± 0.6a | 6.8 ± 0.2a |
| | ZS | 53.9 ± 2.5b | 43.7 ± 1.0b | 36.8 ± 1.8bc | 6.5 ± 0.1b | 4.6 ± 0.4b | 4.3 ± 0.9b |
| | ZPro-Ca | 53.5 ± 1.4b | 41.6 ± 2.8b | 31.6 ± 2.8c | 9.2 ± 0.2a | 9.6 ± 0.2a | 6.9 ± 0.1a |
| | ZPro-Ca+S | 51.8 ± 3.5b | 44.2 ± 2.1b | 41.7 ± 0.4b | 9.6 ± 0.6a | 8.2 ± 0.5a | 6.8 ± 0.3a |
| 21 | XCK | 83.3 ± 1.5a | 58.1 ± 5.4a | 49.0 ± 0.8a | 12.6 ± 0.2a | 11.4 ± 0.7a | 7.8 ± 0.6a |
| | XS | 73.7 ± 1.2bc | 48.0 ± 2.6ab | 38.0 ± 1.0c | 10.5 ± 0.2b | 6.4 ± 0.5c | 5.0 ± 0.6b |
| | XPro-Ca | 79.2 ± 1.6ab | 43.5 ± 2.8b | 44.8 ± 0.7b | 12.7 ± 0.3a | 9.8 ± 0.8ab | 7.6 ± 0.2a |
| | XPro-Ca+S | 71.0 ± 2.6c | 41.7 ± 4.8b | 41.5 ± 2.1bc | 12.1 ± 0.7a | 9.2 ± 0.5b | 7.4 ± 0.2a |
| | ZCK | 76.0 ± 0.6a | 64.0 ± 6.1a | 56.6 ± 1.0a | 10.5 ± 0.3a | 10.0 ± 1.0a | 7.9 ± 0.2a |
| | ZS | 68.2 ± 2.2b | 55.7 ± 0.3a | 41.5 ± 1.5b | 7.8 ± 0.3b | 5.6 ± 0.5b | 5.2 ± 0.2c |
| | ZPro-Ca | 73.3 ± 3.2ab | 57.7 ± 3.3a | 51.1 ± 2.1a | 10.9 ± 0.8a | 11.1 ± 0.3a | 8.3 ± 0.2a |
| | ZPro-Ca+S | 72.3 ± 0.9ab | 41.5 ± 4.3b | 50.5 ± 2.6a | 9.7 ± 0.3a | 8.9 ± 0.6a | 7.3 ± 0.1b |
| 28 | XCK | 86.8 ± 1.9a | 71.6 ± 3.9a | 51.6 ± 2.2a | 16.0 ± 0.1b | 12.1 ± 1.2a | 8.5 ± 0.5a |
| | XS | 83.2 ± 1.5ab | 58.8 ± 2.8bc | 45.0 ± 1.2bc | 11.9 ± 0.5c | 8.2 ± 0.7a | 5.9 ± 0.6b |
| | XPro-Ca | 81.4 ± 1.7b | 62.8 ± 1.3b | 48.1 ± 1.6ab | 17.6 ± 0.5a | 13.2 ± 0.6a | 7.9 ± 0.2a |
| | XPro-Ca+S | 80.5 ± 0.4b | 52.0 ± 1.4c | 42.3 ± 0.6c | 15.3 ± 0.5b | 11.9 ± 2.6a | 7.5 ± 0.0a |
| | ZCK | 88.2 ± 1.4a | 73.5 ± 2.1a | 61.3 ± 0.9a | 14.5 ± 0.5a | 10.6 ± 0.8a | 8.4 ± 0.1b |
| | ZS | 70.5 ± 0.3c | 58.4 ± 1.4c | 44.9 ± 2.9b | 8.9 ± 0.5c | 6.1 ± 0.4b | 5.5 ± 0.2c |
| | ZPro-Ca | 76.9 ± 3.2b | 65.6 ± 2.0b | 57.3 ± 3.5a | 12.4 ± 0.5b | 10.7 ± 0.3a | 8.8 ± 0.3b |
| | ZPro-Ca+S | 74.2 ± 0.7bc | 70.3 ± 1.4ab | 56.0 ± 3.5a | 12.1 ± 0.8b | 9.6 ± 0.2a | 9.7 ± 0.1a |
| 35 | XCK | 86.2 ± 2.6ab | 70.0 ± 3.9a | 58.8 ± 1.0a | 19.8 ± 1.0a | 13.4 ± 0.9a | 9.9 ± 0.5a |
| | XS | 80.3 ± 1.6b | 64.4 ± 0.8ab | 51.3 ± 0.2bc | 15.6 ± 0.4b | 9.5 ± 0.6b | 7.5 ± 0.3b |
| | XPro-Ca | 87.3 ± 1.3a | 63.8 ± 2.4ab | 52.8 ± 0.4b | 20.0 ± 1.1a | 15.8 ± 0.7a | 10.0 ± 0.9a |
| | XPro-Ca+S | 82.0 ± 2.0ab | 55.8 ± 4.9b | 49.2 ± 1.4c | 18.9 ± 0.3a | 13.4 ± 1.1a | 8.3 ± 0.0ab |
| | ZCK | 90.0 ± 0.3a | 84.0 ± 2.0a | 75.8 ± 2.0a | 14.9 ± 0.8a | 10.3 ± 0.2b | 9.3 ± 0.1a |
| | ZS | 80.2 ± 0.6b | 70.3 ± 6.6b | 61.0 ± 5.1b | 12.3 ± 0.5b | 9.5 ± 0.3b | 6.8 ± 0.2b |
| | ZPro-Ca | 76.3 ± 0.5bc | 68.8 ± 1.9b | 61.9 ± 2.8b | 15.8 ± 0.5a | 13.3 ± 0.9a | 9.8 ± 0.5a |
| | ZPro-Ca+S | 75.1 ± 2.4c | 69.6 ± 2.1b | 63.8 ± 0.2b | 15.5 ± 0.3a | 10.6 ± 0.2b | 9.3 ± 0.1a |

**Note:**
Values described are the means ± SE ($n = 3$). Different letters denote significant difference from Duncan's LSD test ($p < 0.05$).

**Table 2 Effects of Pro-Ca on root length of rice main stem and leaf area per stem of rice main stem, first tiller, and second tiller stems at the tillering stage under salt stress.**

| Time/d | Treatment | Root length/cm | Leaf area per stem/cm$^2$ | | |
|---|---|---|---|---|---|
| | | | 0 | I | II |
| 7 | XCK | 32.2 ± 0.2a | 5,370.7 ± 726.7ab | 1,817.1 ± 158.7b | 1,297.3 ± 249.1b |
| | XS | 16.7 ± 2.1c | 4,305.7 ± 149.7b | 974.4 ± 196.4c | 863.1 ± 169.3b |
| | XPro-Ca | 32.3 ± 0.6a | 6,739.5 ± 461.0a | 2,828.0 ± 171.5a | 995.5 ± 141.1b |
| | XPro-Ca+S | 25.4 ± 0.6b | 6,116.9 ± 293.6a | 2,589.2 ± 117.2a | 1,857.3 ± 57.1a |
| | ZCK | 28.0 ± 0.4a | 4,316.7 ± 701.1a | 2,139.5 ± 320.7a | 1,812.1 ± 29.7a |
| | ZS | 20.2 ± 0.4c | 2,503.7 ± 181.5b | 573.0 ± 47.9b | 628.0 ± 114.9b |
| | ZPro-Ca | 28.3 ± 0.7a | 4,958.6 ± 249.8a | 2,751.1 ± 330.2a | 1,735.9 ± 327.2a |
| | ZPro-Ca+S | 24.1 ± 0.6b | 4,556.0 ± 449.6a | 2,459.9 ± 198.9a | 1,909.4 ± 183.4a |
| 14 | XCK | 34.0 ± 0.7a | 8,768.8 ± 265.8ab | 4,157.0 ± 342.1a | 2,850.5 ± 19.0a |
| | XS | 23.3 ± 1.0d | 5,765.6 ± 533.7c | 2,625.8 ± 151.3b | 1,610.0 ± 138.4b |
| | XPro-Ca | 30.4 ± 0.1b | 9,411.7 ± 502.4a | 3,800.7 ± 95.0a | 2,544.2 ± 169.6a |
| | XPro-Ca+S | 26.5 ± 0.9c | 7,524.2 ± 54.7b | 3,416.4 ± 357.4ab | 2,659.9 ± 94.3a |
| | ZCK | 31.6 ± 0.5a | 4,542.4 ± 699.9bc | 3,800.3 ± 49.7b | 2,476.3 ± 155.0b |
| | ZS | 24.0 ± 0.5c | 3,191.6 ± 310.9c | 1,884.3 ± 108.6c | 1,335.7 ± 146.5c |
| | ZPro-Ca | 30.4 ± 0.2a | 7,599.9 ± 557.3a | 5,933.6 ± 537.9a | 4,297.4 ± 297.2a |
| | ZPro-Ca+S | 29.1 ± 0.2b | 5,725.1 ± 198.5b | 3,424.3 ± 163.0b | 2,854.3 ± 261.0b |
| 21 | XCK | 36.9 ± 0.9a | 13,073.6 ± 31.9b | 9,086.0 ± 1,632.7a | 3,487.7 ± 242.7b |
| | XS | 26.3 ± 0.3c | 8,518.4 ± 824.1c | 4,431.7 ± 473.8b | 2,343.2 ± 156.9c |
| | XPro-Ca | 35.1 ± 0.7a | 16,377.4 ± 1,268.1a | 9,106.6 ± 655.3a | 4,243.2 ± 107.5ab |
| | XPro-Ca+S | 31.1 ± 0.4b | 13,157.8 ± 1,026.7b | 7,390.5 ± 261.9ab | 4,539.4 ± 405.0a |
| | ZCK | 31.3 ± 0.3a | 11,510.1 ± 169.6ab | 8,480.1 ± 214.0b | 4,263.7 ± 158.3b |
| | ZS | 25.9 ± 0.3c | 6,699.3 ± 469.0c | 3,349.3 ± 104.3c | 2,419.2 ± 134.3c |
| | ZPro-Ca | 31.9 ± 0.4a | 12,499.1 ± 591.7a | 9,743.8 ± 430.3a | 5,790.4 ± 431.2a |
| | ZPro-Ca+S | 28.1 ± 0.5b | 10,685.9 ± 115.6b | 8,323.6 ± 267.9b | 5,174.6 ± 300.1ab |
| 28 | XCK | 38.9 ± 0.6a | 19,356.5 ± 214.9a | 10,053.9 ± 1,078.9a | 4,877.8 ± 232.1a |
| | XS | 28.9 ± 0.5c | 11,285.3 ± 1,893.9b | 4,555.5 ± 186.5c | 3,383.9 ± 246.4b |
| | XPro-Ca | 35.5 ± 0.2b | 21,134.8 ± 1,004.4a | 11,031.7 ± 344.0a | 5,339.8 ± 348.8a |
| | XPro-Ca+S | 35.7 ± 0.6b | 18,674.2 ± 442.4a | 7,893.0 ± 501.8b | 4,676.2 ± 44.4a |
| | ZCK | 34.3 ± 1.0a | 13,441.2 ± 333.7a | 8,997.6 ± 286.3b | 6,685.3 ± 605.7a |
| | ZS | 27.6 ± 0.6b | 7,290.5 ± 219.1c | 4,316.1 ± 337.7c | 3,482.6 ± 238.6b |
| | ZPro-Ca | 34.9 ± 0.4a | 12,883.3 ± 750.9a | 9,499.3 ± 462.4b | 7,186.9 ± 142.7a |
| | ZPro-Ca+S | 34.8 ± 0.3a | 10,919.9 ± 694.5b | 10,803.5 ± 466.6a | 6,794.2 ± 293.9a |
| 35 | XCK | 41.9 ± 0.7a | 22,136.7 ± 1754.2a | 15,401.6 ± 3,314.5ab | 7,314.4 ± 1013.4a |
| | XS | 30.0 ± 0.4b | 14,410.6 ± 860.6b | 6,125.3 ± 374.9c | 4,154.9 ± 161.7a |
| | XPro-Ca | 39.7 ± 0.7a | 22,637.8 ± 649.4a | 18,446.2 ± 1,148.1a | 7,497.9 ± 1,618.8a |
| | XPro-Ca+S | 41.0 ± 1.0a | 20,386.9 ± 589.8a | 11,051.6 ± 981.7bc | 5,948.2 ± 318.4a |
| | ZCK | 36.4 ± 0.6a | 17,411.4 ± 454.4a | 11,695.5 ± 378.4b | 9,364.4 ± 773.9a |
| | ZS | 30.7 ± 0.3b | 10,297.9 ± 287.1d | 8,110.1 ± 300.1c | 4,612.9 ± 130.9b |
| | ZPro-Ca | 38.3 ± 1.2a | 14,462.8 ± 419.4b | 12,506.7 ± 509.8b | 8,107.0 ± 497.7a |
| | ZPro-Ca+S | 37.8 ± 0.5a | 12,335.2 ± 361.4c | 14,148.4 ± 387.9a | 7,825.9 ± 929.4a |

**Note:**
Values are the means ± SE ($n$ = 3). Different letters denote significant difference from Duncan's LSD test ($p < 0.05$).

**Table 3 Effects of Pro-Ca on dry weight per stem of rice main stem, first tiller, and second tiller stems and root dry weight of rice main stem at the tillering stage under salt stress.**

| Time/ d | Treatment | Dry weight per stem/ g | | | Root dry weight/ g |
|---|---|---|---|---|---|
| | | 0 | I | II | |
| 7 | XCK | 0.470 ± 0.017a | 0.168 ± 0.004a | 0.103 ± 0.005b | 0.195 ± 0.007a |
| | XS | 0.340 ± 0.015b | 0.062 ± 0.011b | 0.065 ± 0.006c | 0.118 ± 0.007d |
| | XPro-Ca | 0.478 ± 0.045a | 0.178 ± 0.003a | 0.133 ± 0.005a | 0.170 ± 0.005b |
| | XPro-Ca+S | 0.466 ± 0.011a | 0.156 ± 0.015a | 0.094 ± 0.012b | 0.146 ± 0.004c |
| | ZCK | 0.426 ± 0.053a | 0.168 ± 0.023b | 0.129 ± 0.004a | 0.154 ± 0.005b |
| | ZS | 0.202 ± 0.005b | 0.058 ± 0.004c | 0.045 ± 0.008b | 0.098 ± 0.002c |
| | ZPro-Ca | 0.392 ± 0.011a | 0.212 ± 0.009a | 0.143 ± 0.012a | 0.189 ± 0.007a |
| | ZPro-Ca+S | 0.365 ± 0.006a | 0.159 ± 0.008b | 0.138 ± 0.008a | 0.139 ± 0.013b |
| 14 | XCK | 0.626 ± 0.020a | 0.295 ± 0.019b | 0.198 ± 0.002a | 0.268 ± 0.011ab |
| | XS | 0.392 ± 0.040b | 0.188 ± 0.004b | 0.124 ± 0.006c | 0.140 ± 0.000c |
| | XPro-Ca | 0.573 ± 0.028a | 0.440 ± 0.074a | 0.206 ± 0.014a | 0.318 ± 0.053a |
| | XPro-Ca+S | 0.532 ± 0.016a | 0.239 ± 0.012b | 0.168 ± 0.002b | 0.191 ± 0.004bc |
| | ZCK | 0.464 ± 0.017a | 0.352 ± 0.033a | 0.191 ± 0.013b | 0.235 ± 0.008b |
| | ZS | 0.305 ± 0.027b | 0.148 ± 0.012c | 0.101 ± 0.004c | 0.112 ± 0.001c |
| | ZPro-Ca | 0.552 ± 0.026a | 0.415 ± 0.032a | 0.262 ± 0.010a | 0.311 ± 0.010a |
| | ZPro-Ca+S | 0.468 ± 0.050a | 0.245 ± 0.007b | 0.236 ± 0.024ab | 0.229 ± 0.009b |
| 21 | XCK | 1.145 ± 0.066a | 0.382 ± 0.003c | 0.199 ± 0.003b | 0.405 ± 0.020b |
| | XS | 0.638 ± 0.065b | 0.345 ± 0.029c | 0.182 ± 0.008b | 0.266 ± 0.007d |
| | XPro-Ca | 1.194 ± 0.078a | 0.579 ± 0.024a | 0.250 ± 0.020a | 0.266 ± 0.007d |
| | XPro-Ca+S | 1.249 ± 0.030a | 0.460 ± 0.018b | 0.259 ± 0.016a | 0.266 ± 0.007d |
| | ZCK | 0.988 ± 0.049a | 0.650 ± 0.019a | 0.291 ± 0.012b | 0.331 ± 0.021c |
| | ZS | 0.565 ± 0.005c | 0.256 ± 0.014b | 0.222 ± 0.020c | 0.206 ± 0.009d |
| | ZPro-Ca | 1.008 ± 0.062a | 0.681 ± 0.047a | 0.367 ± 0.012a | 0.596 ± 0.013a |
| | ZPro-Ca+S | 0.791 ± 0.037b | 0.598 ± 0.031a | 0.359 ± 0.018a | 0.493 ± 0.035b |
| 28 | XCK | 2.160 ± 0.134a | 0.943 ± 0.041b | 0.344 ± 0.009a | 1.469 ± 0.073a |
| | XS | 0.896 ± 0.023c | 0.617 ± 0.033c | 0.224 ± 0.002b | 0.428 ± 0.046d |
| | XPro-Ca | 1.828 ± 0.070b | 1.373 ± 0.117a | 0.326 ± 0.030a | 1.278 ± 0.013b |
| | XPro-Ca+S | 1.571 ± 0.045b | 0.633 ± 0.029c | 0.346 ± 0.022a | 0.601 ± 0.048c |
| | ZCK | 1.387 ± 0.121a | 0.767 ± 0.051b | 0.376 ± 0.013b | 0.954 ± 0.012a |
| | ZS | 0.654 ± 0.019b | 0.400 ± 0.008c | 0.278 ± 0.011b | 0.284 ± 0.011d |
| | ZPro-Ca | 1.467 ± 0.114a | 0.948 ± 0.047a | 0.689 ± 0.030a | 0.695 ± 0.019b |
| | ZPro-Ca+S | 1.235 ± 0.126a | 0.908 ± 0.076ab | 0.824 ± 0.077a | 0.631 ± 0.020c |
| 35 | XCK | 2.817 ± 0.131a | 1.765 ± 0.056a | 0.446 ± 0.042ab | 1.938 ± 0.081ab |
| | XS | 1.053 ± 0.009c | 0.725 ± 0.011b | 0.350 ± 0.013b | 1.014 ± 0.075c |
| | XPro-Ca | 2.802 ± 0.072a | 1.547 ± 0.258a | 0.603 ± 0.086a | 2.153 ± 0.219a |
| | XPro-Ca+S | 2.502 ± 0.099b | 1.432 ± 0.039a | 0.449 ± 0.036ab | 1.639 ± 0.037b |
| | ZCK | 2.243 ± 0.104ab | 1.265 ± 0.139b | 0.544 ± 0.076b | 1.446 ± 0.059a |
| | ZS | 1.575 ± 0.054c | 0.881 ± 0.056c | 0.429 ± 0.012b | 0.551 ± 0.019c |
| | ZPro-Ca | 2.321 ± 0.144a | 1.539 ± 0.024a | 1.341 ± 0.052a | 0.842 ± 0.079b |
| | ZPro-Ca+S | 1.915 ± 0.121bc | 1.421 ± 0.018ab | 1.212 ± 0.017a | 0.767 ± 0.035b |

**Note:**
Values are the means ± SE ($n$ = 3). Different letters denote significant difference from Duncan's LSD test ($p < 0.05$).

application of Pro-Ca under NaCl stress increased the number of tillers by 14.32% to 59.99% and 42.86% to 100.04% in Xiangliangyou900 and Huanghuazhan, respectively, from the 7th to the 35th days. Compared with S treatment, the number of main stem leaves of both rice varieties increased significantly in Pro-Ca+S treatment, where the number of main stem leaves of Xiangliangyou900 increased by 4.16%, 23.33%, 3.03%, 23.53%, and 6.25%, respectively, and the number of main stem leaves of Huanghuazhan increased by 11.11%, 10.00%, 15.62%, 19.44% and 17.78%, respectively (Figs. 1E and 1F).

As shown in Table 1, compared with the control, foliar spraying Pro-Ca significantly reduced the plant height of the two rice varieties, in which the plant height of Xiangliangyou900 was reduced by 5.00–15.00% and Huanghuazhan's plant height was reduced by 3.51–20.42% from the 7th to the 35th days. The spraying of Pro-Ca reduced the first tiller length of Xiangliangyou900 and Huanghuazhan by 8.77–30.02% and 9.90–26.17% from the 7th to the 35th days, respectively, and the second tiller length of the two varieties was reduced by 6.91–22.69% and 6.52–38.96% (Table 1). Compared with the S treatment, foliar spraying Pro-Ca under NaCl stress significantly alleviated the stem base width of each tiller position in both rice, in which the main stem basal width, first tiller basal width and second tiller basal width of Xiangliangyou900 were increased by 15.24–44.07%, 30.87–45.30%, and 10.58–48.32%, respectively, from the 7th to the 35th days, the stem base width of each tiller position increased by 23.84–47.56%, 11.19–76.13%, and 36.28–76.96%, respectively (Table 1). Compared with S treatment, foliar spraying Pro-Ca under NaCl stress increased the leaf area of main stem, first tiller leaf area and second tiller leaf area of Xianglaingyou900 by 30.50–65.47%, 30.11–165.73%, and 38.19–115.18% from the 7th to the 35th days, respectively, and the leaf area of each tiller position of Huanghuazhan increased by 19.78–81.97%, 74.45–329.32%, and 69.65–204.06%, respectively (Table 2). The root lengths of Xiangliangyou900 and Huanghuazhan under Pro-Ca+S treatment increased by 13.43% to 52.29% and 8.37% to 26.09%, respectively, from the 7th to the 35th days compared with that of S treatment, and the differences were significant (Table 2). As shown in Table 3, compared with NaCl, the main stem dry weight, first tiller dry weight and second tiller dry weight of Pro-Ca+S treatment of Xiangliangyou900 increased by 35.75–137.61%, 2.51–152.69%, and 28.29–54.31%, respectively, and the dry weights of each tiller position of Huanghuazhan increased by 21.60–88.90%, 61.33–174.87%, and 61.22–205.01%, respectively. Compared with the S treatment, foliar spraying of Pro-Ca alleviated the suppression of below-ground biomass by NaCl stress, and the root dry weight of Xiangliangyou900 increased by 23.64–61.74% and that of Huanghuazhan increased by 39.22–139.70% from the 7th to the 35th days (Table 3).

### Effect of salt stress on antioxidant enzymes in rice leaves at each tiller position at tillering stage and regulation by Pro-Ca

Compared with the control, the SOD activity of the main stem leaves of Xiangliangyou900 increased by 6.88% to 31.25% from the 7th to the 35th days after salt treatment, and the SOD activity of the main stem leaves of Huanghuazhan decreased by 10.86% and 9.81% on the 7th and 14th days, respectively, and increased by 13.00% to 24.32% from the 21st to the

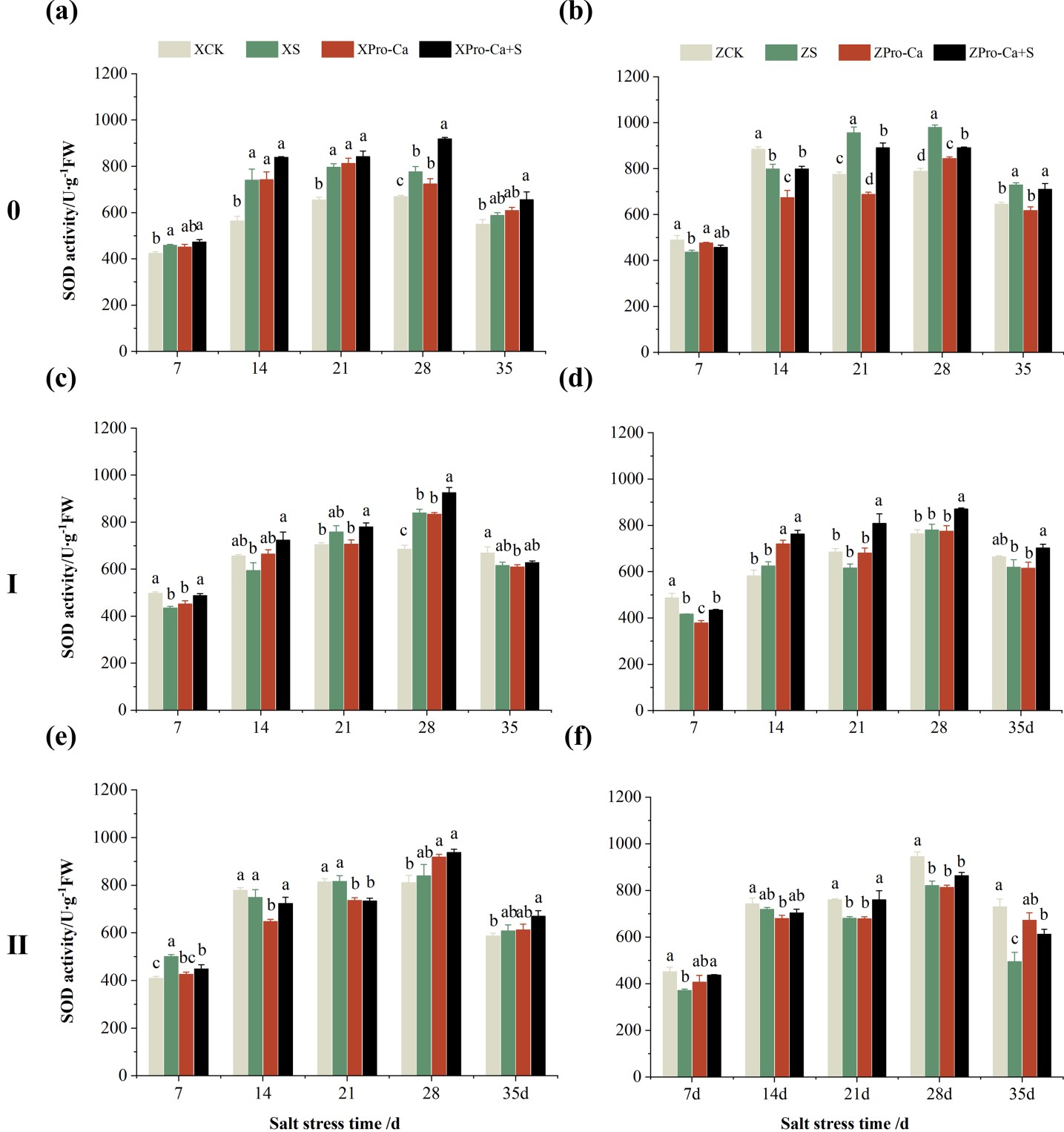

**Figure 2 Effect of Pro-Ca on SOD activity of rice main stem (A, B), first tiller (C, D), and second tiller (E, F) leaves under salt stress.** The different letters are significant differences according to Duncan's new multiple range test ($p < 0.05$) based on one-way ANOVA. Xiangliangyou900: XCK (distilled water), XS (0.3% NaCl), XPro-Ca (100 mg·L$^{-1}$ Pro-Ca), XPro-Ca+S (100 mg·L$^{-1}$ Pro-Ca + 0.3% NaCl), Huanghuazhan: ZCK (distilled water), ZS (0.3% NaCl), ZPro-Ca (100 mg·L$^{-1}$ Pro-Ca), ZPro-Ca+S (100 mg·L$^{-1}$ Pro-Ca + 0.3% NaCl).

35th days (Figs. 2A and 2B). The SOD activity of the first tiller leaves of Xiangliangyou900 decreased by 12.55% and 9.39% on the 7th and 14th days after salt treatment, respectively, and that of the first tiller leaves of Huanghuazhan decreased by 14.48% on the 7th day, and did not show any significant difference compared with the control in the following days (Figs. 2C and 2D). Compared with the control, the SOD activity of the second tiller leaves of Xiangliangyou900 was significantly increased by 22.51% on the 7th day but did not change significantly from the 14th to the 35th days after NaCl treatment, however, the SOD activity of the second tiller leaves of Huanghuazhan decreased by 3.17% to 32.22% from the 7th to the 35th days (Figs. 2E and 2F).

As can be seen from Figs. 3A and 3B, compared with the control, salt stress reduced the CAT activity of main stem leaves of Xiangliangyou900 by 9.03% to 9.63% from the 7th to the 28th days, and that of Huanghuazhan main stem leaves by 1.89% to 16.99% from the 14th to the 35th days. Meanwhile, as shown in Fig. 3, NaCl stress reduced the CAT activity of the second tiller leaves of Xiangliangyou900 by 0.50% to 32.15% from the 7th to the 35th days, respectively, and that of Huanghuazhan's first tiller leaves by 3.90% to 8.54% from the 21st to 35th days after salt stress, and that of the second tiller leaves by 0.78% to 5.62%.

NaCl stress reduced the POD activity of main stem leaves of Xiangliangyou900 by 2.83% and 13.67% on the 7th and 14th days, respectively (Fig. 4A), and that of Huanghuazhan by 13.01% and 6.55%, respectively, compared with the control (Fig. 4B). In addition, salt stress reduced the POD activity of the first tiller leaves of Xiangliangyou900 by 10.81% and 8.82% on the 7th and 14th days, and increased it by 5.12% and 24.40% on the 21st and 28th days, respectively, compared with the CK (Fig. 4C). The POD activity of the first tiller leaves of Huanghuazhan was reduced by 5.71% to 10.71% from the 7th to the 21st days and increased by 77.45% and 66.59% at the 28th and 35th days, respectively (Fig. 4D). Under salt stress, the POD activities of the second tiller leaves of Xiangliangyou900 and Huanghuazhan were reduced by 3.69–44.75% and 4.67–26.11%, respectively, compared with the control from the 7th to the 35th days (Figs. 4E and 4F).

The results in Figs. 5A and 5B showed that NaCl stress increased increased the APX activity of main stem leaves of Xiangliangyou900 by 9.02% to 89.09% from the 14th to the 35th days, and decreased the APX activity of Huanghuazhan's by 56.24% and 7.31% on the 7th and 14th days, respectively, and increased it by 8.55% to 16.30% from the 21st to 35th days. Compared with the CK treatment, the APX activity of the first tiller leaves of Xiangliangyou900 and Huanghuazhan decreased by 12.80–54.26% and 3.38–50.04%, respectively, from the 7th to the 35th days after NaCl stress (Figs. 5C and 5D), and that of the second tiller leaves of the two varieties decreased by 5.62–34.90% and 2.74–33.33%, respectively (Figs. 5E and 5F).

Under Pro-Ca+S treatment, the SOD activity of main stem leaves of Xiangliangyou900 significantly increased by 3.08% to 18.45% from the 7th to the 35th days, and that of Huanghuazhan was only increased by 4.76% and 0.09% on the 7th and 14th days, respectively (Figs. 2A and 2B). The SOD activity of the first tiller leaves of both varieties were increased by 1.93% to 21.78% and 4.14% to 31.35% from the 7th to the 35th days after spraying with Pro-Ca compared to S treatment, respectively (Figs. 2C and 2D). In addition, it can be seen from Figs. 2E and 2F that under salt stress, spraying Pro-Ca increased the
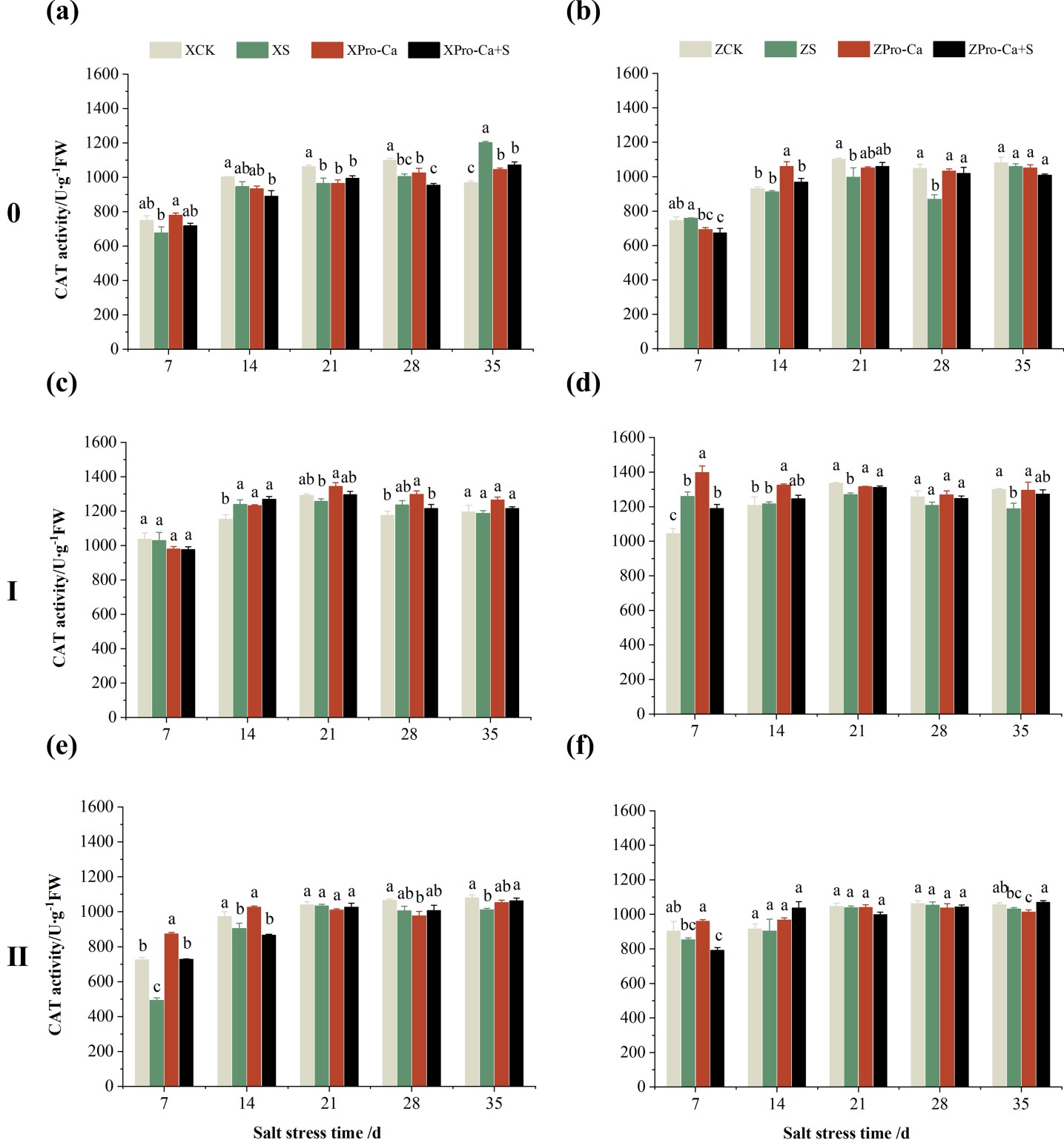

**Figure 3 Effect of Pro-Ca on CAT activity of rice main stem (A, B), first tiller (C, D), and second tiller (E, F) leaves under salt stress.** The different letters are significant differences according to Duncan's new multiple range test ($p < 0.05$) based on one-way ANOVA. Xiangliangyou900: XCK (distilled water), XS (0.3% NaCl), XPro-Ca (100 mg·L$^{-1}$ Pro-Ca), XPro-Ca+S (100 mg·L$^{-1}$ Pro-Ca + 0.3% NaCl), Huanghuazhan: ZCK (distilled water), ZS (0.3% NaCl), ZPro-Ca (100 mg·L$^{-1}$ Pro-Ca), ZPro-Ca+S (100 mg·L$^{-1}$ Pro-Ca + 0.3% NaCl).
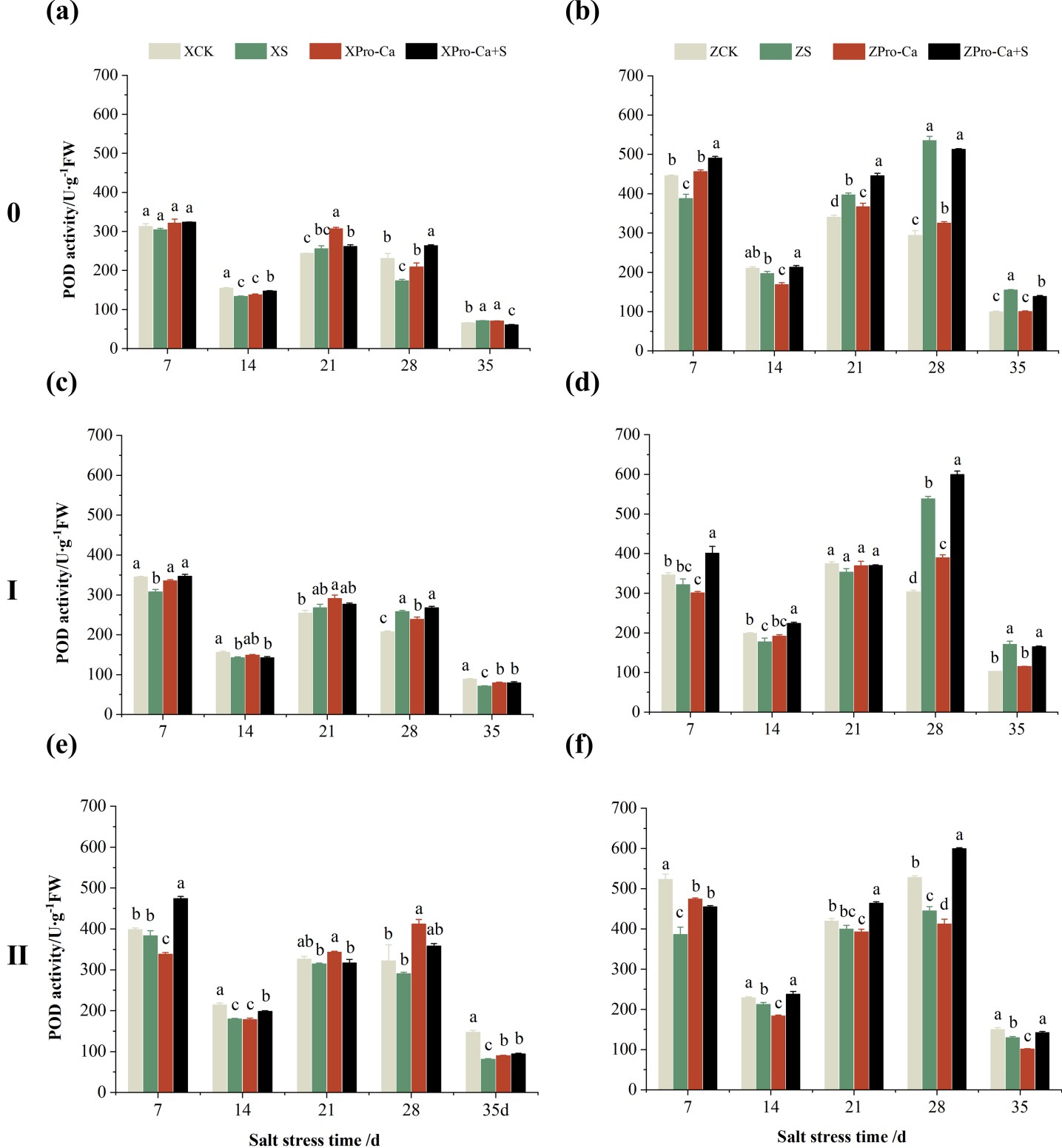

**Figure 4 Effect of Pro-Ca on POD activity of rice main stem (A, B), first tiller (C, D), and second tiller (E, F) leaves under salt stress.** The different letters are significant differences according to Duncan's new multiple range test ($p < 0.05$) based on one-way ANOVA. Xiangliangyou900: XCK (distilled water), XS (0.3% NaCl), XPro-Ca (100 mg·L$^{-1}$ Pro-Ca), XPro-Ca+S (100 mg·L$^{-1}$ Pro-Ca + 0.3% NaCl), Huanghuazhan: ZCK (distilled water), ZS (0.3% NaCl), ZPro-Ca (100 mg·L$^{-1}$ Pro-Ca), ZPro-Ca+S (100 mg·L$^{-1}$ Pro-Ca + 0.3% NaCl).

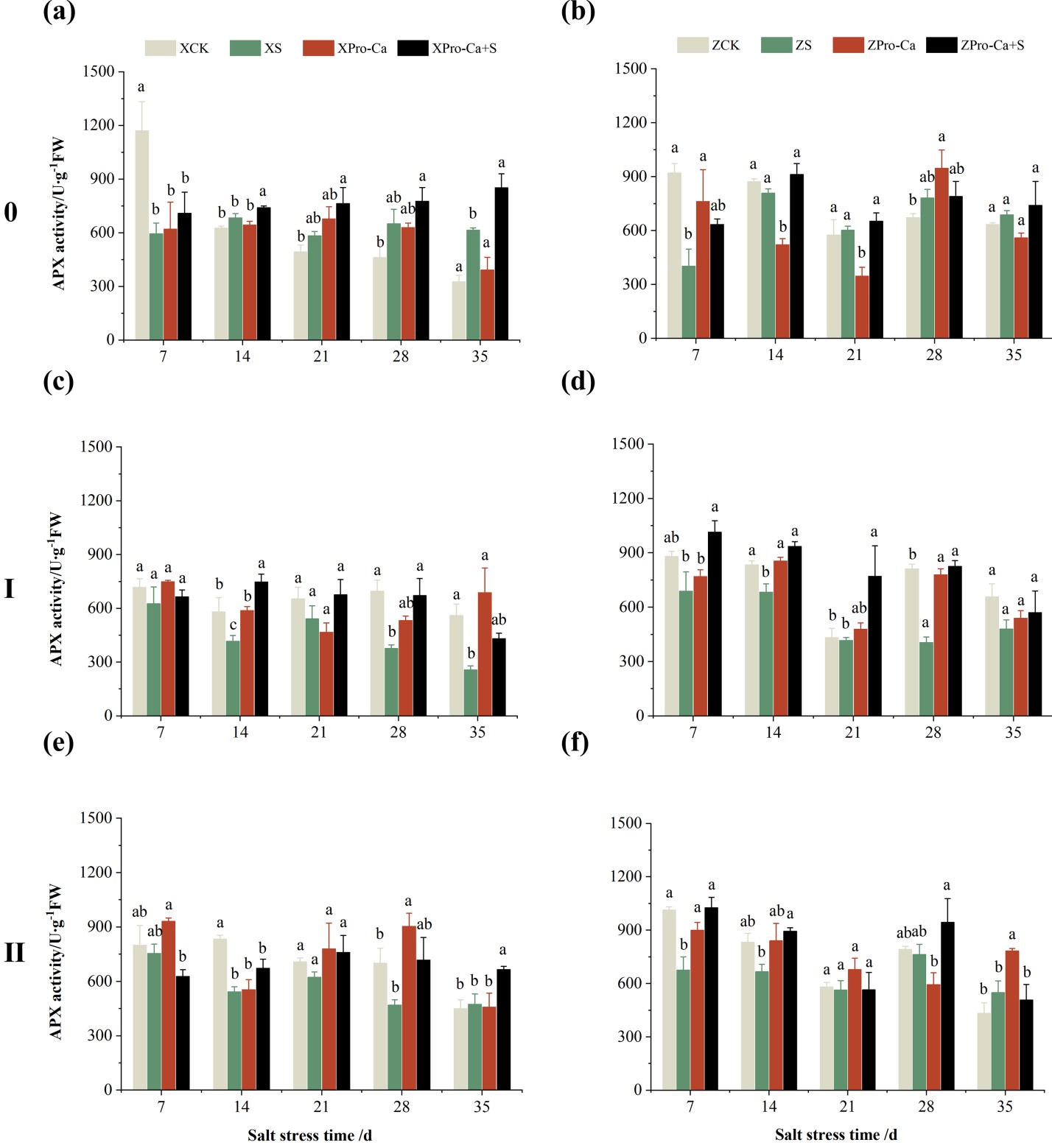

**Figure 5 Effect of Pro-Ca on APX activity of rice main stem (A, B), first tiller (C, D), and second tiller (E, F) leaves under salt stress.** The different letters are significant differences according to Duncan's new multiple range test ($p < 0.05$) based on one-way ANOVA. Xiangliangyou900: XCK (distilled water), XS (0.3% NaCl), XPro-Ca (100 mg·L$^{-1}$ Pro-Ca), XPro-Ca+S (100 mg·L$^{-1}$ Pro-Ca + 0.3% NaCl), Huanghuazhan: ZCK (distilled water), ZS (0.3% NaCl), ZPro-Ca (100 mg·L$^{-1}$ Pro-Ca), ZPro-Ca+S (100 mg·L$^{-1}$ Pro-Ca + 0.3% NaCl).

SOD activity of the second tiller leaves of Xiangliangyou900 by 11.64% and 10.13% on the 28th and 35th days, respectively, and that of Huanghuazhan by 5.16–23.86% from the 21st to the 35th days.

Compared with salt stress, spraying Pro-Ca before salt stress had no significant effect on the CAT activity of leaves of all tillers of Xiangliangyou900, in which, it increased the CAT activity of leaves of the main stem of Huanghuazhan by 6.18–17.17% from the 14th to the 28th days, respectively. Compared with the S treatment, spraying Pro-Ca increased the CAT activity of the first tiller leaves of Huanghuazhan by 2.37–7.12% from the 14th to the 28th days, respectively (Fig. 3).

As can be seen in Fig. 4, the POD activity of the main stem leaves of Xiangtwoyou900 under Pro-Ca+S treatment increased by 2.38% to 51.79% from the 7th to the 28th days, and that of the main stem leaves of Huanghuazhan increased by 8.27% to 26.63% from the 7th to the 21st days, as compared with that of S treatment. The POD activities of the first tiller and second tiller leaves of Xiangliangyou900 under Pro-Ca+S treatment increased by 0.45% to 12.84% and 0.83% to 23.74%, respectively, from the 7th to the 35th days (Figs. 4C and 4E). In addition, the spraying of Pro-Ca increased the POD activity of the first tiller leaves of Huanghuazhan under salt stress by 4.84% to 26.07% from the 7th to the 28th days after salt stress, and that of the second tiller leaves by 9.82% to 34.71% from the 7th to the 35th days (Figs. 4D and 4F).

Compared with NaCl stress alone, spraying Pro-Ca under NaCl stress increased the APX activity of main stem leaves of Xiangliangyou900 by 8.48% to 38.46% from the 7th to the 35th days (Fig. 5A). The APX activity of main stem leaves of Huanghuazhan increased by 1.10% to 57.35% from the 7th to the 35th days, but not significantly (Fig. 5B). Spraying Pro-Ca under NaCl stress increased the APX activity of the first tiller leaves of Xiangliangyou900 and Huanghuazhan by 6.15–79.71% and 19.02–103.47%, respectively, from the 7th to the 35th days (Figs. 5C and 5D). The APX activity of the second tiller leaves of Xiangliangyou900 was increased by 2.38–40.31% from the 14th to the 35th days (Fig. 5E). The activity of the second tiller leaves of Huanghuazhan was increased by 0.15–51.94% from the 7th to the 28th days (Fig. 5F).

### Effect of salt stress on antioxidant enzymes in rice stems at each tiller position at tillering stage and regulation by Pro-Ca

Compared with the CK treatment, the SOD activity of the main stem of Xiangliangyou900 decreased by 7.64% to 30.56% from the 7th to the 35th days after NaCl treatment, and that of the main stem of Huanghuazhan decreased by 1.43% to 13.72% from the 21st to the 35th days (Figs. 6A and 6B). The SOD activity in the stem of the first tiller of Xiangliangyou900 decreased by 3.81% to 11.08% from the 7th to the 35th days after NaCl treatment, and that in the stem of the second tiller decreased by 1.88% to 32.36% from the 7th to the 28th days (Figs. 6C and 6E). NaCl stress reduced the SOD activity in the stem of the first tiller of Huanghuazhan by 4.98% to 52.21% from the 21st to the 35th days, and reduced the SOD activity in the stem of the second tiller by 3.27% to 26.61% from the 7th to the 35th days, respectively (Figs. 6D and 6F).

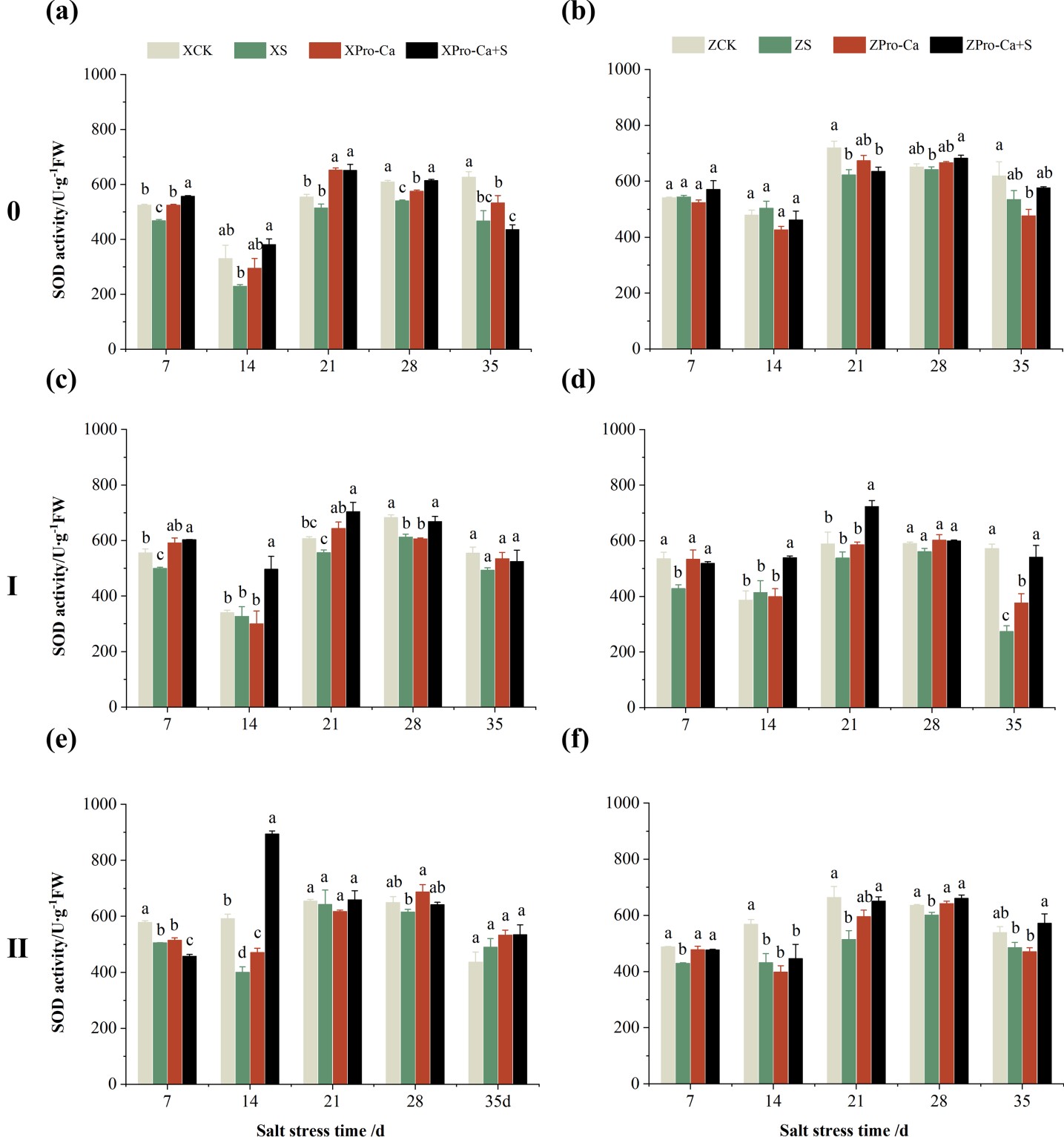

**Figure 6 Effect of Pro-Ca on SOD activity of rice main stem (A, B), first tiller (C, D), and second tiller (E, F) stems under salt stress.** The different letters are significant differences according to Duncan's new multiple range test ($p < 0.05$) based on one-way ANOVA. Xiangliangyou900: XCK (distilled water), XS (0.3% NaCl), XPro-Ca (100 mg·L$^{-1}$ Pro-Ca), XPro-Ca+S (100 mg·L$^{-1}$ Pro-Ca + 0.3% NaCl), Huanghuazhan: ZCK (distilled water), ZS (0.3% NaCl), ZPro-Ca (100 mg·L$^{-1}$ Pro-Ca), ZPro-Ca+S (100 mg·L$^{-1}$ Pro-Ca + 0.3% NaCl).

Figure 7 shows that NaCl stress reduced the main stem CAT activity of Xiangliangyou900 by 5.85% to 25.60% from the 7th to the 28th days. The CAT activity of Huanghuazhan decreased by 6.67% and 26.94% on the 14th and 35th days, respectively, and increased but not significantly at the 7th, 21st, and 28th days. The CAT activity of the first tiller stems of Xiangliangyou900 decreased by 3.10% to 26.99% from the 21st to the 35th days, and that of Huanghuazhan decreased by 3.14%, 3.27%, and 45.44% at the 7th, 21st, and 35th, respectively (Figs. 7C and 7D). NaCl stress reduced the CAT activity in the stem of the second tiller of Xiangliangyou900 by 8.69% to 38.70% from the 7th to the 28th days after NaCl stress, and that of Huanghuazhan by 10.25% to 37.52% from the 7th to the 35th days (Figs. 7E and 7F).

Compared with their respective CKs, the POD activity of main stem of Xiangliangyou900 under NaCl stress increased by 5.02% to 44.88% from the 7th to the 35th days, and that of Huanghuazhan increased by 7.39% to 11.37% from the 7th to the 21st days, and decreased by 18.49% and 35.13% at the 28th and 35th, respectively (Figs. 8A and 8B). The POD activity of the first tiller stems of Xiangliangyou900 decreased by 5.51% to 45.94% in 7–35 days under salt stress (Fig. 8C). The POD activity in the stem of the first tiller of Huanghuazhan decreased by 2.64% to 10.67% from the 7th to the 28th days, and increased by 4.11% at the 28th day (Fig. 8D). Compared with the control, salt stress reduced the POD activity in the stem of the second tiller by 2.48% to 25.92% and 2.68% to 25.68% in the stem of the second tiller of Xiangliangyou900 and Huanghuazhan, respectively, from the 7th to the 35th days (Figs. 8E and 8F).

The main stem APX activity of Xiangliangyou900 decreased by 12.12% to 39.98% from the 7th to the 35th days after NaCl stress, and that of Huanghuazhan decreased by 2.65% to 34.28%, respectively (Figs. 9A and 9B). The APX activity in the stem of the first and second tillers of both varieties decreased by 2.57% to 49.67% and 5.29% to 64.09%, 21.47% to 63.03% and 21.35% to 34.96%, respectively, from the 7th to the 35th days after salt stress (Fig. 9).

Compared with the S treatment, spraying Pro-Ca under salt stress increased the SOD activity of the main stem of Xiangliangyou900 by 13.66% to 65.88% from the 7th to the 28th days, and that of the main stem of Huanghuazhan by 2.05% to 7.96% from the 21st to the 35th days (Figs. 6A and 6B), the SOD activities in the stem of the first tiller increased by 6.34–51.90% and 6.91–97.69% from the 7th to the 35th days in Xiangliangyou900 and Huanghuazhan, respectively (Figs. 6C and 6D). Spraying Pro-Ca under salt stress increased the SOD activity of the second tiller stems of Xiangliangyou900 by 2.68% to 123.51% from the 14th to the 35th days, and that of the second tiller stems of Huanghuazhan by 3.27% to 26.61% from the 7th to the 35th days (Figs. 6E and 6F).

Compared with the S treatment, the CAT activities of main stems sprayed with Pro-Ca Xiangliangyou900 and Huanghuazhan under salt stress increased by 5.72% to 92.16% and 6.04% to 40.09% from the 7th to the 35th days, respectively (Figs. 7A and 7B). Foliar spraying of Pro-Ca also effectively increased the CAT activity in the first tiller stems of two rice varieties under salt stress, which was increased by 0.83–30.00% at the 7–28 days in Xiangliangyou900 and by 3.59–95.57% at the 7–35 days in Huanghuazhan (Figs. 7C and 7D). CAT activity in the second tiller stems of both varieties was increased by 0.93% to

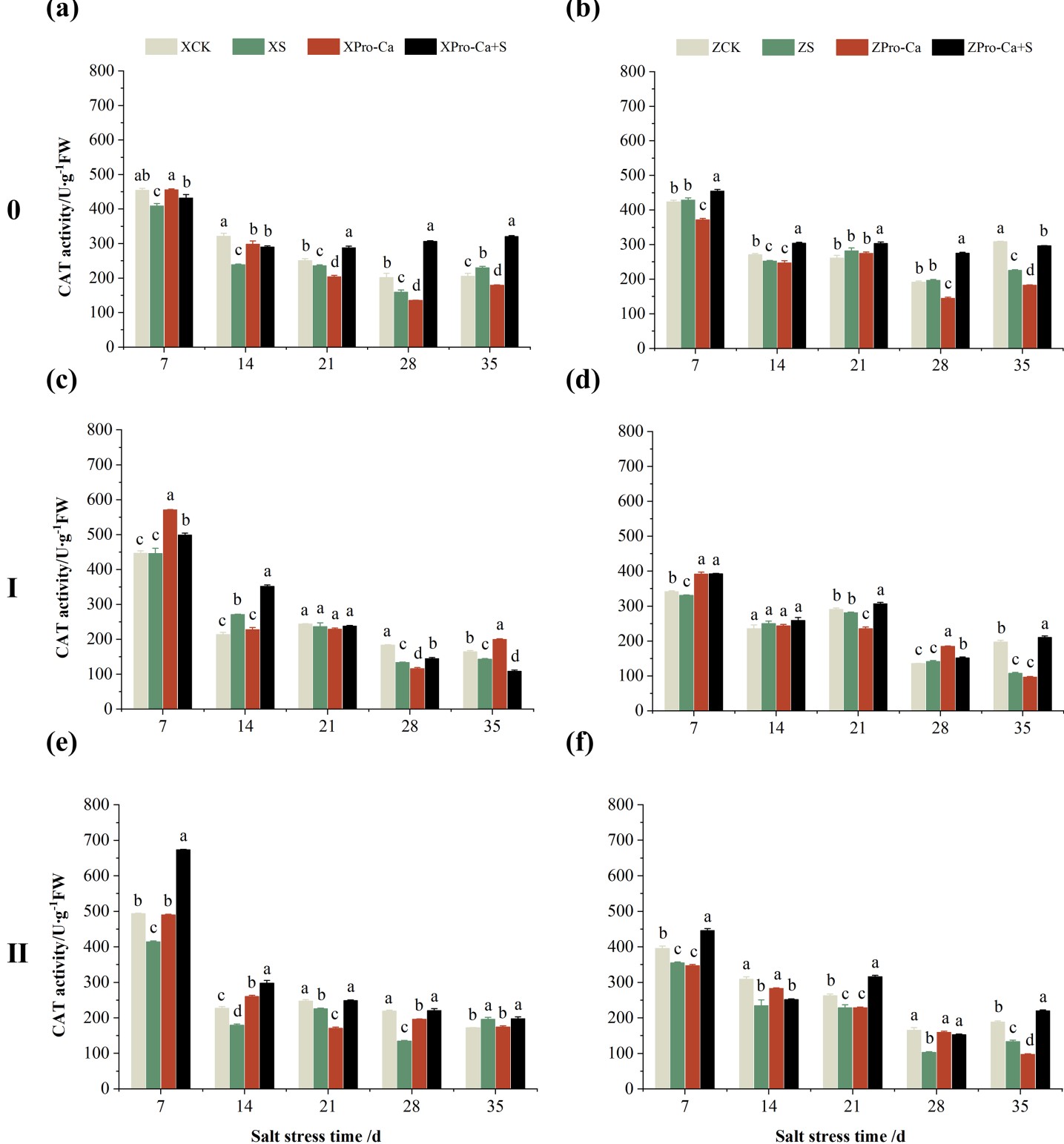

**Figure 7 Effect of Pro-Ca on CAT activity of rice main stem (A, B), first tiller (C, D), and second tiller (E, F) stems under salt stress.** The different letters are significant differences according to Duncan's new multiple range test ($p < 0.05$) based on one-way ANOVA. Xiangliangyou900: XCK (distilled water), XS (0.3% NaCl), XPro-Ca (100 mg·L$^{-1}$ Pro-Ca), XPro-Ca+S (100 mg·L$^{-1}$ Pro-Ca + 0.3% NaCl), Huanghuazhan: ZCK (distilled water), ZS (0.3% NaCl), ZPro-Ca (100 mg·L$^{-1}$ Pro-Ca), ZPro-Ca+S (100 mg·L$^{-1}$ Pro-Ca + 0.3% NaCl).

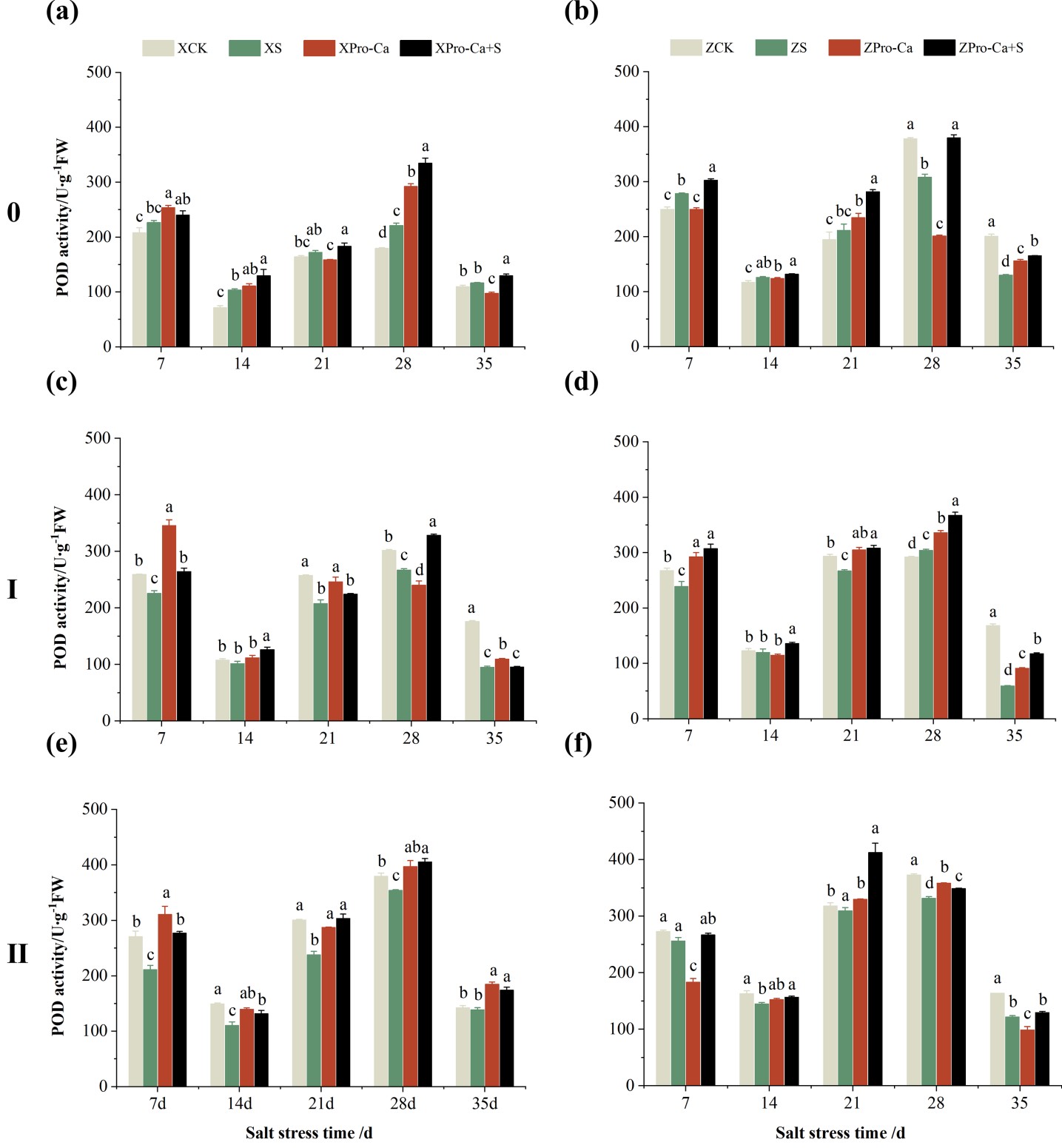

**Figure 8 Effect of Pro-Ca on POD activity of rice main stem (A, B), first tiller (C, D), and second tiller (E, F) stems under salt stress.** The different letters are significant differences according to Duncan's new multiple range test ($p < 0.05$) based on one-way ANOVA. Xiangliangyou900: XCK (distilled water), XS (0.3% NaCl), XPro-Ca (100 mg·L$^{-1}$ Pro-Ca), XPro-Ca+S (100 mg·L$^{-1}$ Pro-Ca + 0.3% NaCl), Huanghuazhan: ZCK (distilled water), ZS (0.3% NaCl), ZPro-Ca (100 mg·L$^{-1}$ Pro-Ca), ZPro-Ca+S (100 mg·L$^{-1}$ Pro-Ca + 0.3% NaCl).

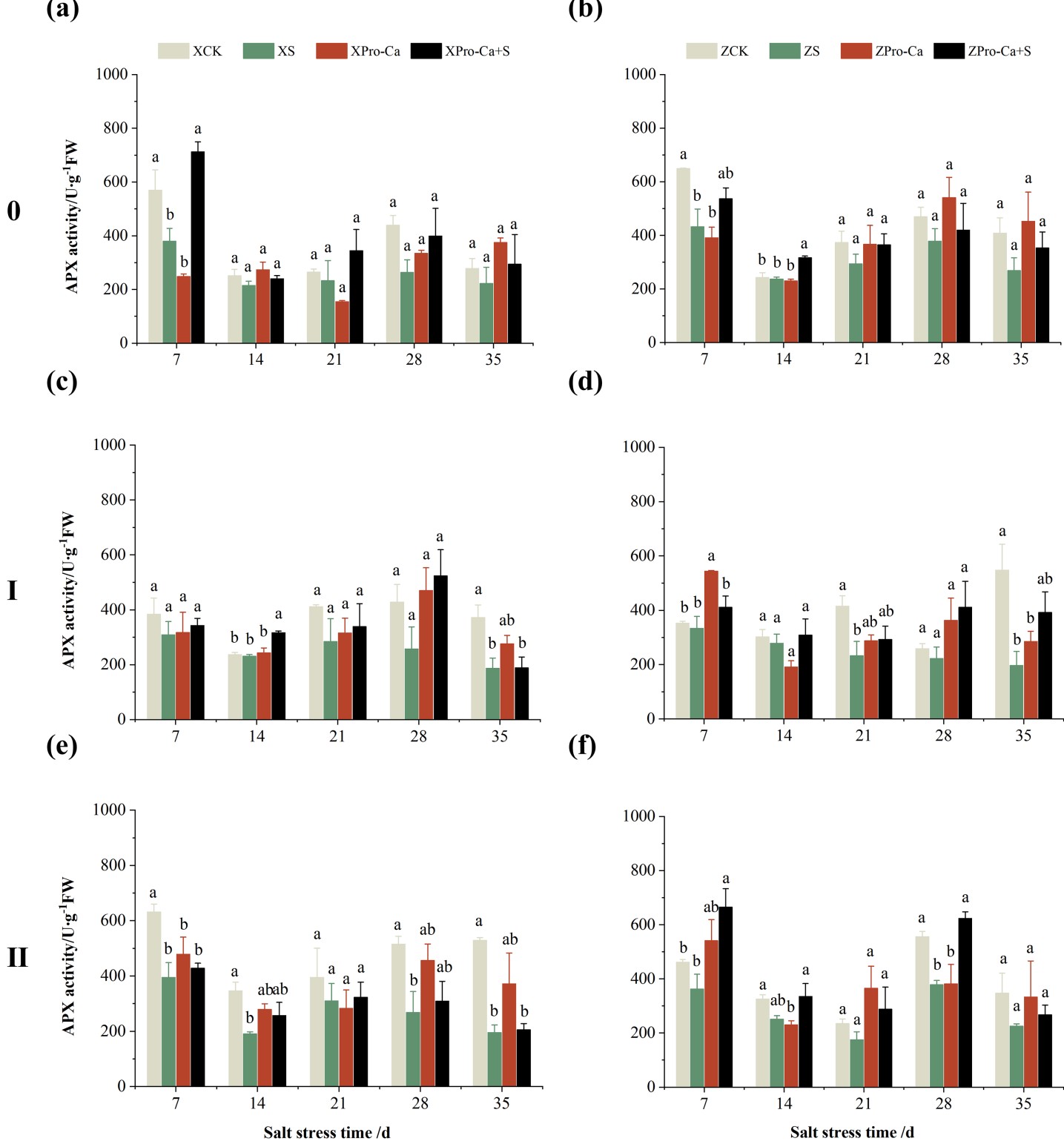

**Figure 9 Effect of Pro-Ca on APX activity of rice main stem (A, B), first tiller (C, D), and second tiller (E, F) stems under salt stress.** The different letters are significant differences according to Duncan's new multiple range test ($p < 0.05$) based on one-way ANOVA. Xiangliangyou900: XCK (distilled water), XS (0.3% NaCl), XPro-Ca (100 mg·L$^{-1}$ Pro-Ca), XPro-Ca+S (100 mg·L$^{-1}$ Pro-Ca + 0.3% NaCl), Huanghuazhan: ZCK (distilled water), ZS (0.3% NaCl), ZPro-Ca (100 mg·L$^{-1}$ Pro-Ca), ZPro-Ca+S (100 mg·L$^{-1}$ Pro-Ca + 0.3% NaCl).

66.28% and 7.57% to 65.13% under Pro-Ca+S treatment compared to S, respectively (Figs. 7E and 7F).

Compared with the S treatment alone, spraying Pro-Ca under salt stress increased the main stem POD activity by 5.89% to 51.39% and 4.76% to 33.32% from the 7th to the 35th days in Xiangliangyou900 and Huanghuazhan, respectively. The POD activity in the first and second tiller stems of both varieties increased by 0.18% to 24.07% and 13.49% to 98.03%, 14.51% to 31.29% and 4.15% to 33.22%, respectively, from the 7th to the 35th days (Fig. 8).

Compared with salt stress alone, spraying Pro-Ca under NaCl stress increased the main stem APX activity of Xiangliangyou900 by 11.43% to 87.63% from the 7th to the 35th days, and that of Huanghuazhan by 10.98% to 31.58%, respectively (Figs. 9A and 9B). It increased the APX activity of the first and second tiller stems of both varieties by 0.85% to 103.18% and 10.74% to 99.20%, 4.11% to 34.69% and 18.49% to 83.24%, respectively, from the 7th to the 35th days (Fig. 9).

## Effect of salt stress on membrane damage index in rice leaves at each tiller position at tillering stage and regulation by Pro-Ca

Salt stress significantly increased the MDA content of leaves of two rice varieties compared with the control (Fig. 10). Among them, the MDA contents of main stem, first tiller and second tiller leaves of Xiangliangyou900 increased by 13.04–54.61%, 0.58–79.40%, and 3.33–39.59%, respectively, from the 7th to the 35th days after salt stress (Figs. 10A, 10C, and 10E), the MDA content of the leaves of each tiller position of Huanghuazhan increased by 10.62% to 127.93%, 13.25% to 75.94%, and 6.01% to 64.67%, respectively, from the 7th to the 35th days after salt stress (Figs. 10B, 10D, and 10F). In addition, as seen in Fig. 11, salt stress increased the $H_2O_2$ content of the main stem leaves of Xiangliangyou900 and Huanghuazhan by 4.57–38.51% and 3.88–21.84% at 7–35 days, and the $H_2O_2$ content of the first tiller leaves of two kinds of rice by 5.24–39.80% and 0.44–41.06%. The $H_2O_2$ content of the second tiller leaves of the two varieties was increased by 16.46–28.88% and 8.37–58.75% at 7–35 days, respectively (Fig. 11).

Compared with S treatment, foliar spraying of Pro-Ca before salt stress decreased the MDA content of main stem leaves of Xiangliangyou900 and Huanghuazhan by 0.53–27.59% and 8.65–25.00%, respectively (Figs. 10A and 10B), and that of the first tiller leaves of the two varieties under the same conditions by 17.24–26.80% and 2.39–29.12%, respectively (Figs. 10C and 10D), from the 7th to the 35th days. The MDA contents of second tiller leaves of the two varieties under the same conditions were decreased by 0.09–34.98% and 1.60–26.93%, respectively (Figs. 10E and 10F). Spraying Pro-Ca before salt stress reduced the $H_2O_2$ content of main stem leaves of Xiangliangyou900 and Huanghuazhan by 0.63–22.87% and 5.91–42.28%, respectively, from the 7th to the 35th days (Figs. 11A and 11B). Spraying Pro-Ca reduced the $H_2O_2$ content of the first tiller leaves of Xiangliangyou900 by 4.03–31.42% and that of Huanghuazhan by 7.56–26.04%, respectively at 7–35 days (Figs. 11C and 11D), and reduced the $H_2O_2$ content of the second tiller leaves of both rice varieties by 5.49–29.10% and 8.23–34.97% (Figs. 11E and 11F). The staining test revealed that there were more spots on the leaves of both rice varieties under

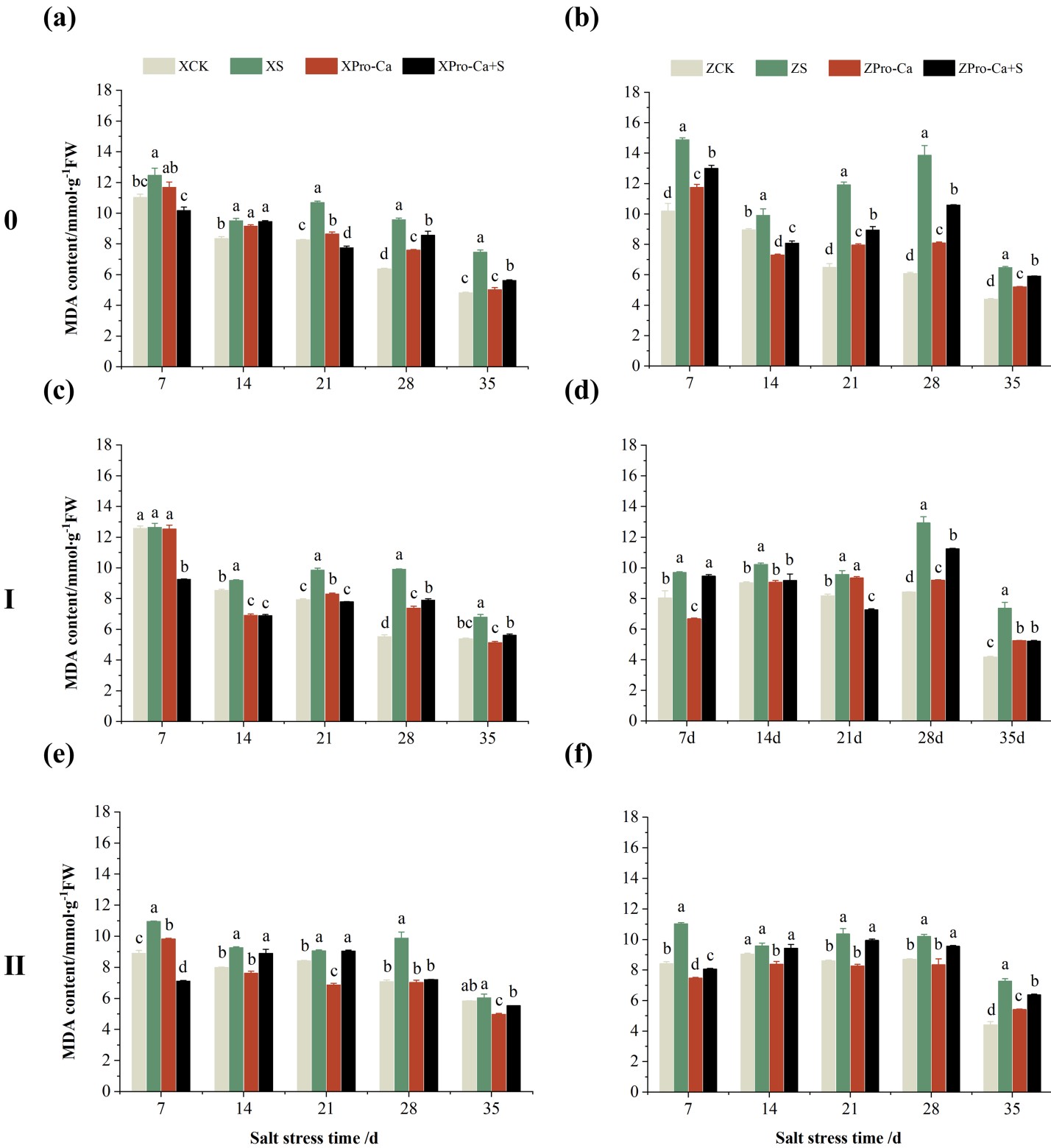

**Figure 10 Effect of Pro-Ca on MDA content of rice main stem (A, B), first tiller (C, D), and second tiller (E, F) leaves under salt stress.** The different letters are significant differences according to Duncan's new multiple range test ($p < 0.05$) based on one-way ANOVA. Xiangliangyou900: XCK (distilled water), XS (0.3% NaCl), XPro-Ca (100 mg·L$^{-1}$ Pro-Ca), XPro-Ca+S (100 mg·L$^{-1}$ Pro-Ca + 0.3% NaCl), Huanghuazhan: ZCK (distilled water), ZS (0.3% NaCl), ZPro-Ca (100 mg·L$^{-1}$ Pro-Ca), ZPro-Ca+S (100 mg·L$^{-1}$ Pro-Ca + 0.3% NaCl).

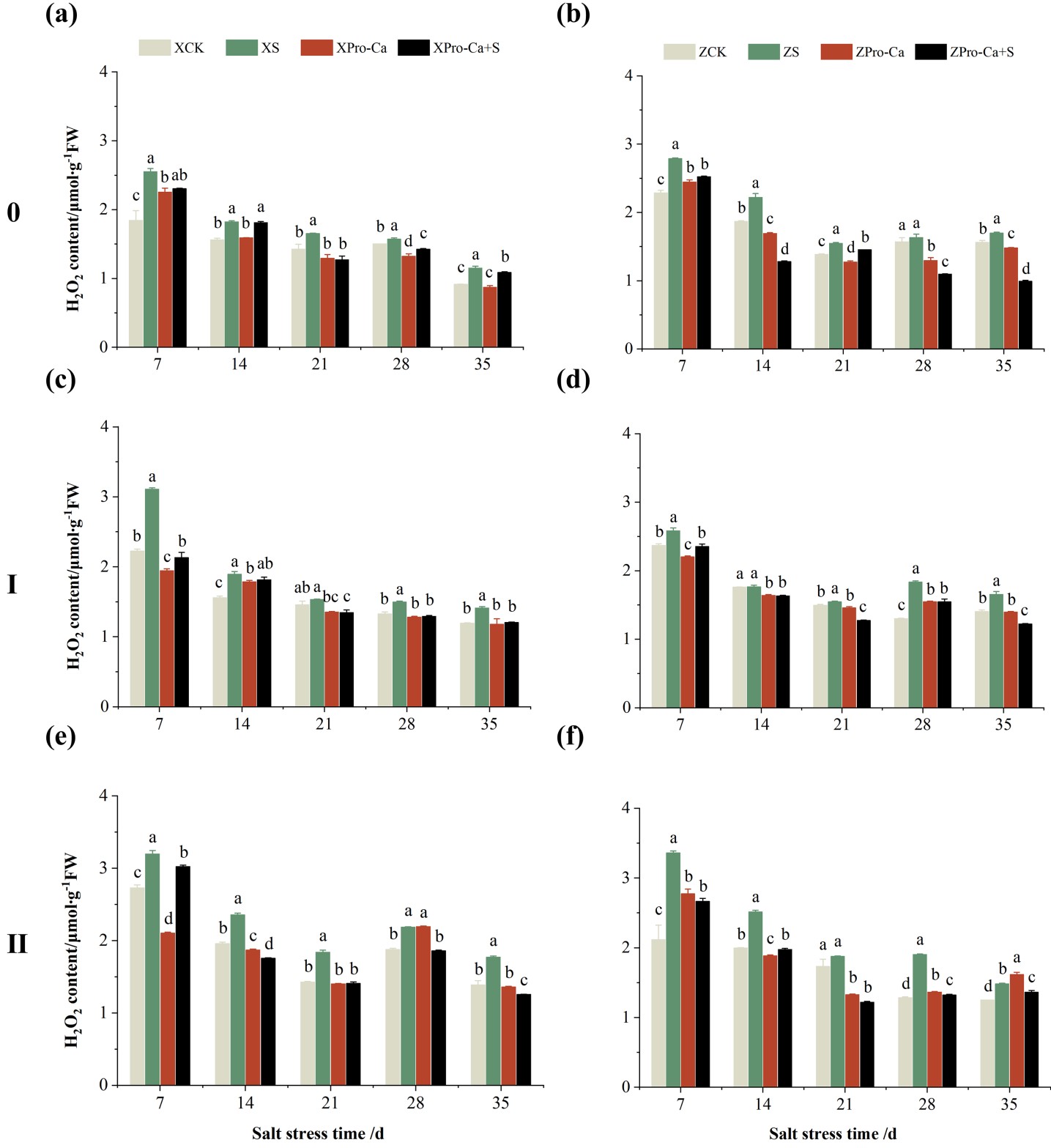

**Figure 11 Effect of Pro-Ca on H₂O₂ content of rice main stem (A, B), first tiller (C, D), and second tiller (E, F) leaves under salt stress.** The different letters are significant differences according to Duncan's new multiple range test ($p < 0.05$) based on one-way ANOVA. Xiangliangyou900: XCK (distilled water), XS (0.3% NaCl), XPro-Ca (100 mg·L⁻¹ Pro-Ca), XPro-Ca+S (100 mg·L⁻¹ Pro-Ca + 0.3% NaCl), Huanghuazhan: ZCK (distilled water), ZS (0.3% NaCl), ZPro-Ca (100 mg·L⁻¹ Pro-Ca), ZPro-Ca+S (100 mg·L⁻¹ Pro-Ca + 0.3% NaCl).

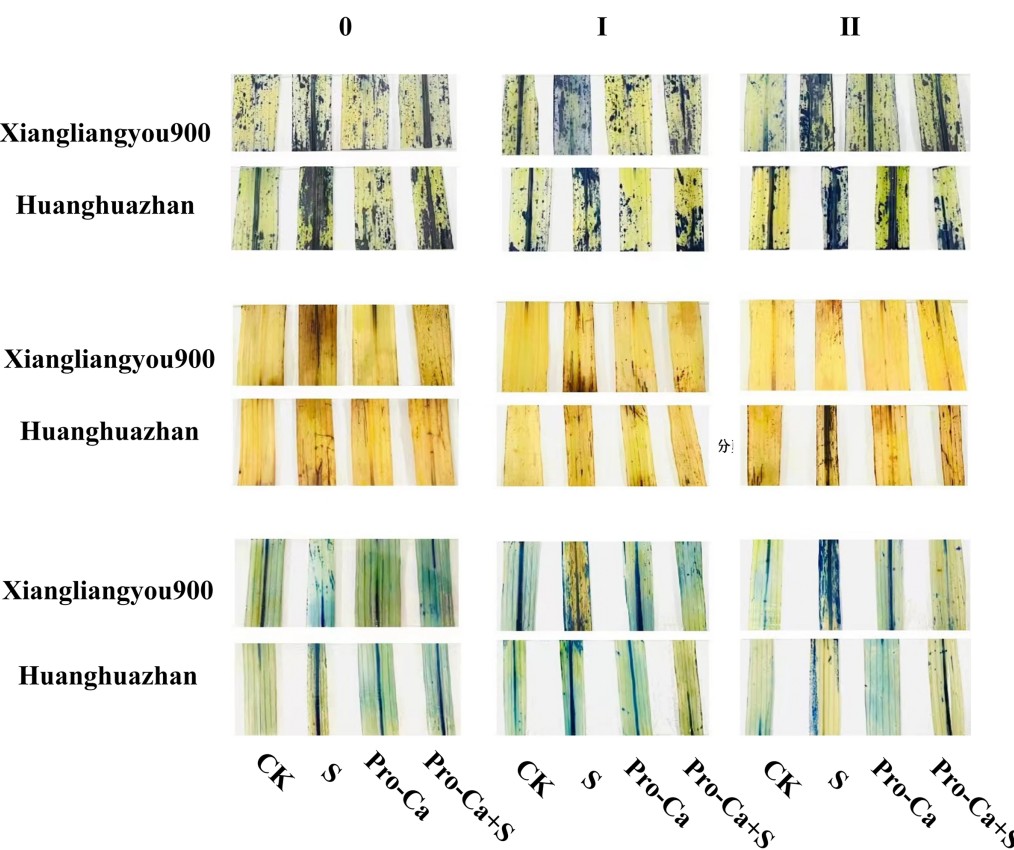

**Figure 12 Effect of Pro-Ca on $O_2^-$, $H_2O_2$ content and cellular activity of rice main stem leaves at each tiller position under salt stress.** From left to right, the three columns represent the tiller stem, the first node position, and the second tiller position. Each image shows the following treatments applied to the plants from left to right: CK (distilled water), S (0.3% NaCl), Pro-Ca (100 mg·L$^{-1}$ Pro-Ca), Pro-Ca+S (100 mg·L$^{-1}$ Pro-Ca + 0.3% NaCl). The spot area represents the degree of stress, and the larger staining area indicates that the more severe stress of leaves.

salt stress, and the spots on the tiller leaves were larger in area and darker in color. In contrast, the area of spots on the leaves of the treatments sprayed with Pro-Ca before salt stress decreased and became lighter in color (Fig. 12).

## Effect of salt stress on membrane damage index in rice stems at each tiller position at tillering stage and regulation by Pro-Ca

Compared with the control, NaCl stress increased the MDA content of the main stem of Xiangliangyou900 and Huanghuazhan by 14.85–115.59% and 39.38–95.94%, respectively (Figs. 13A and 13B), and that of the first tiller stems of Xiangliangyou900 and Huanghuazhan by 4.98–90.72% and 9.55–72.13%, respectively, from the 7th to the 35th days after salt stress (Figs. 13C and 13D), and the MDA content of the second tiller stems of the two varieties increased by 22.21–156.61% and 7.98–61.68%, respectively (Figs. 13E and 13F). NaCl stress at 0.3% also significantly increased the $H_2O_2$ content in the stems of both rice varieties (Fig. 14). Among them, salt stress increased the $H_2O_2$ content of the main stem of Xiangliangyou900 by 1.28% to 6.12% from the 7th to the 35th days, and

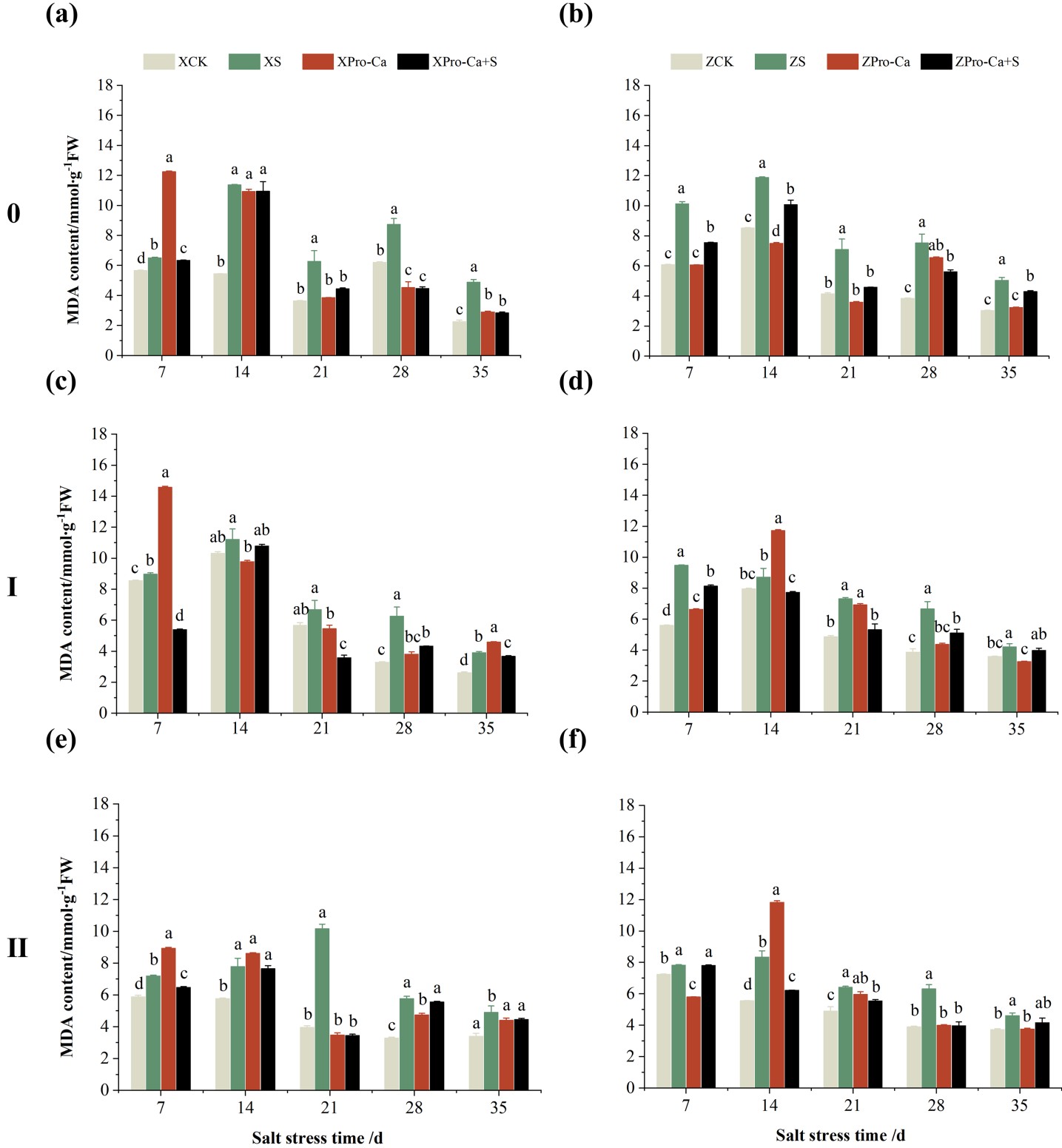

**Figure 13 Effect of Pro-Ca on MDA content of rice main stem (A, B), first tiller (C, D), and second tiller (E, F) stems under salt stress.** The different letters are significant differences according to Duncan's new multiple range test ($p < 0.05$) based on one-way ANOVA. Xiangliangyou900: XCK (distilled water), XS (0.3% NaCl), XPro-Ca (100 mg·L$^{-1}$ Pro-Ca), XPro-Ca+S (100 mg·L$^{-1}$ Pro-Ca + 0.3% NaCl), Huanghuazhan: ZCK (distilled water), ZS (0.3% NaCl), ZPro-Ca (100 mg·L$^{-1}$ Pro-Ca), ZPro-Ca+S (100 mg·L$^{-1}$ Pro-Ca + 0.3% NaCl).

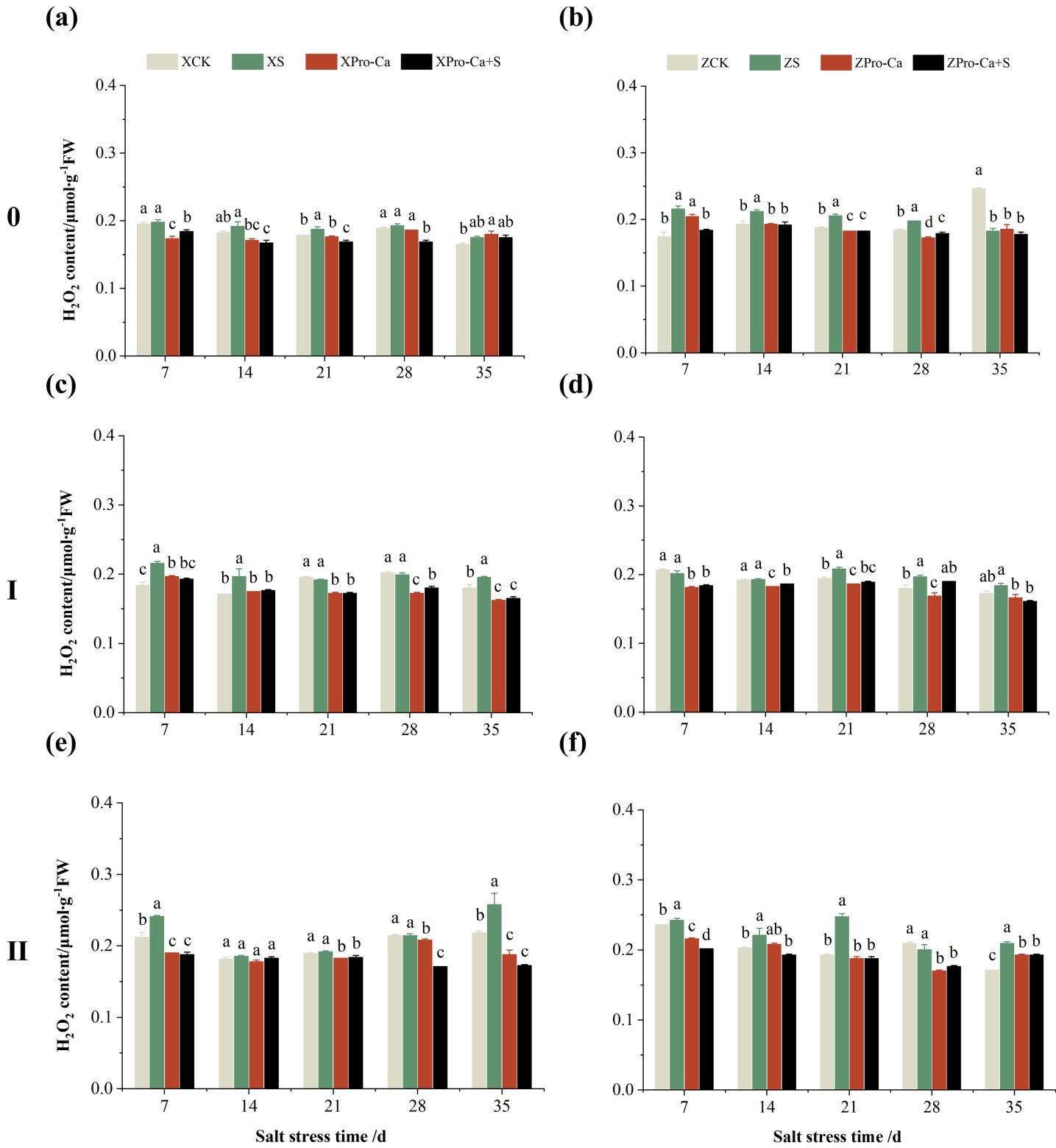

**Figure 14 Effect of Pro-Ca on H₂O₂ content of rice main stem (A, B), first tiller (C, D), and second tiller (E, F) stems under salt stress.** The different letters are significant differences according to Duncan's new multiple range test ($p < 0.05$) based on one-way ANOVA. Xiangliangyou900: XCK (distilled water), XS (0.3% NaCl), XPro-Ca (100 mg·L⁻¹ Pro-Ca), XPro-Ca+S (100 mg·L⁻¹ Pro-Ca + 0.3% NaCl), Huanghuazhan: ZCK (distilled water), ZS (0.3% NaCl), ZPro-Ca (100 mg·L⁻¹ Pro-Ca), ZPro-Ca+S (100 mg·L⁻¹ Pro-Ca + 0.3% NaCl).

increased the $H_2O_2$ content of the main stem of Huanghuazhan by 7.61% to 24.09% from the 7th to the 28th days (Figs. 14A and 14B). Salt stress increased the $H_2O_2$ content of the first tiller stem of Xiangliangyou900 by 17.28%, 14.83%, and 8.49% at the 7th, 14th, and 35th days, respectively, and that of Huanghuazhan by 0.63–9.16% at 14–35 days (Figs. 14C and 14D). In addition, NaCl stress increased the $H_2O_2$ content in the stem of the second tiller of Xiangliangyou900 by 1.37–18.04% from the 7th to the 35th days, and there was no significant change on the 28th day, that of Huanghuazhan increased the content by 2.67% and 28.36% on the at the 7th and 21st days (Figs. 14E and 14F).

Compared with S treatment, spraying Pro-Ca before salt stress caused a significant decrease in MDA content, in which the MDA content of the main stems of Xiangliangyou900 and Huanghuazhan decreased by 2.53% to 48.87% and 14.94% to 35.39%, respectively, from the 7th to the 35th days (Figs. 13A and 13B). The MDA content of the first tiller stems of the two varieties sprayed with Pro-Ca decreased by 3.79–46.45% and 5.39–27.28%, respectively, from the 7th to the 35th days after salt stress (Figs. 13C and 13D), compared with S treatment, spraying Pro-Ca decreased the MDA content of the second tiller stems of Xiangliangyou900 and Huanghuazhan by 1.70–66.09% and 0.32–37.09% from the 7th to the 35th days, respectively (Figs. 13E and 13F). Foliar spraying of Pro-Ca before salt stress reduced $H_2O_2$ in the main stem of Xiangliangyou900 by 7.07–12.62% from the 7th to the 28th days, and that of Huanghuazhan by 2.79–14.74% from the 7th to the 35th days (Figs. 14A and 14B), and reduced the $H_2O_2$ content in the stem of the first tiller of Xiangliangyou900 and Huanghuazhan by 9.58–15.60% and 3.20–12.45% from the 7th to the 35th days respectively (Figs. 14C and 14D). Compared with salt stress, spraying Pro-Ca reduced the $H_2O_2$ content in the stem of the second tiller of the two rice varieties by 1.35–33.05% and 7.92–24.15%, respectively, from the 7th to the 35th days (Figs. 14E and 14F).

### Effect of salt stress on soluble protein content in rice leaves at each tiller position at tillering stage and regulation by Pro-Ca

As can be seen from Fig. 15, the soluble protein content of the main stem leaves of Xiangliangyou900 increased by 0.88% to 2.59% from the 7th to the 28th days after salt stress, and the soluble protein content of the main stem leaves of Huanghuazhan decreased by 1.59% and 2.87% on the 7th and 14th days after salt stress, respectively, and increased by 0.44% to 2.41% from the 21st to 35th d. NaCl stress increased the soluble protein content of the first tiller leaves of Xiangliangyou900 by 0.61% and 5.33% on the 21st and 28th days, respectively, and decreased the soluble protein content of the first tiller leaves of Huanghuazhan by 2.06% to 4.22% from the 7th to the 28th days after NaCl stress (Figs. 15C and 15D). The soluble protein content of the second tiller leaves of Xiangtwoyou900 increased by 3.27% and 0.45% on the 7th and 21st after NaCl stress, and decreased that of Huanghuazhan by 0.31% to 4.63% from the 7th to the 35th days (Figs. 15E and 15F).

Compared with the S treatment, the soluble protein content of the main stem leaves of Xiangliangyou900 increased by 0.21% to 4.67% from the 7th to the 35th days in Pro-Ca+S treatment, and that of the main stem leaves of Huanghuazhan increased by 1.28% to 1.90% from the 7th to the 21st days (Figs. 15A and 15B). In addition, spraying Pro-Ca under salt

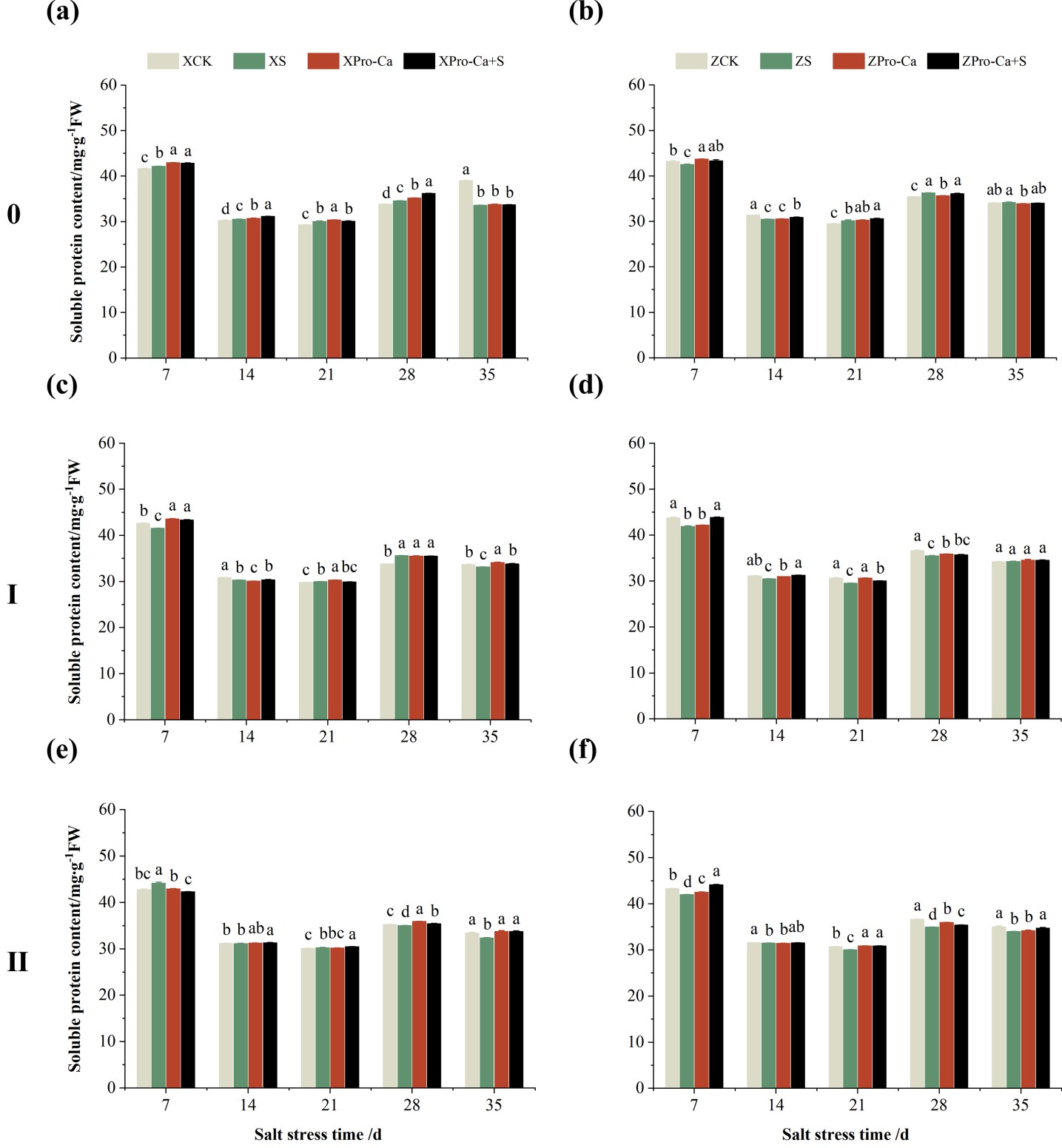

**Figure 15 Effect of Pro-Ca on the soluble protein content of rice main stem (A, B), first tiller (C, D), and second tiller (E, F) leaves under salt stress.** The different letters are significant differences according to Duncan's new multiple range test ($p < 0.05$) based on one-way ANOVA. Xiangliangyou900: XCK (distilled water), XS (0.3% NaCl), XPro-Ca (100 mg·L$^{-1}$ Pro-Ca), XPro-Ca+S (100 mg·L$^{-1}$ Pro-Ca + 0.3% NaCl), Huanghuazhan: ZCK (distilled water), ZS (0.3% NaCl), ZPro-Ca (100 mg·L$^{-1}$ Pro-Ca), ZPro-Ca+S (100 mg·L$^{-1}$ Pro-Ca + 0.3% NaCl).

stress increased the soluble protein content of the first tiller leaves of Xiangliangyou900 by 4.30%, 0.25%, and 1.94% at the 7th, 14th, and 35th, respectively, and that of Huanghuazhan by 0.57% to 4.64% from the 7th to the 35th days (Figs. 15C and 15D). Foliar spraying of Pro-Ca increased the soluble protein content of the second tiller leaves of Xiangliangyou900 by 0.55% to 4.38% from the 14th to the 35th days and Huanghuazhan by 0.16% to 5.15% from the 7th to the 35th days after salt stress (Figs. 15E and 15F).

## Effect of salt stress on soluble protein content in rice stems at each tiller position at tillering stage and regulation by Pro-Ca

As can be seen from the Figs. 16A and 16B, the soluble protein content of the main stem of Xiangliangyou900 increased by 1.92% to 9.30% from the 14th to the 35th days after salt stress, and the soluble protein content of the main stem of Huanghuazhan increased by 1.03% to 7.41% from the 7th to the 28th days. Salt stress reduced the soluble protein content of the first tiller stems of Xiangliangyou900 by 0.31%, 5.92%, and 3.83% at the 7th, 28th, and 35th days, and reduced the soluble protein content of the first tiller stems of Huanghuazhan by 1.48% and 1.12% at the 21st and 35th days, respectively (Figs. 16C and 16D). The soluble protein content of the second tiller stems of Xiangliangyou900 decreased by 0.88% to 12.45% from the 7th to the 35th days after salt stress, and there was no significant difference at the 14th day, while that of the second tiller stems of Huanghuazhan decreased by 4.81% to 12.49% from the 14th to the 35th days (Figs. 16E and 16F).

Spraying Pro-Ca under salt stress conditions increased the soluble protein content of the main stems of Xiangliangyou900 and Huanghuazhan by 0.97–9.13% and 2.05–15.38%, respectively, from the 7th to the 35th days after NaCl stress (Figs. 16A and 16B). Compared with the S treatment, the soluble protein content of the first tiller stems of Xiangliang900 under Pro-Ca+S treatment increased by 0.90% to 14.89% from the 7th to the 35th days, but there was no significant difference at the 21st day, and that of the first tiller stems of Huanghuazhan under Pro-Ca+S treatment increased by 3.89% to 22.64% from the 7th to the 35th days (Figs. 16C and 16D). Foliar spraying of Pro-Ca under NaCl stress increased the soluble protein content of the second tiller stems of Xiangliangyou900 and Huanghuazhan by 8.09% to 26.44% and 2.11% to 17.37%, respectively, from the 7th to the 35th days after salt stress (Figs. 16E and 16F).

## DISCUSSION

Salt stress, as a significant abiotic stressor, can greatly inhibit crop growth and development. It has the ability to trigger the production of ROS, an excess of which can harm cellular functions and even result in cell death. ROS also act as secondary messengers, activating molecular processes in response to environmental stresses. Therefore, maintaining an appropriate level of ROS is crucial for plant survival under stress conditions (Lee et al., 2022). However, excessive salt stress can disrupt the balance of internal ions in cells (Zulfiqar & Ashraf, 2021). High concentrations of salt stress also increase the production of ROS, such as mono-linear oxygen ($^1O_2$), superoxide ($O_2^-$), hydrogen peroxide ($H_2O_2$), and hydroxyl radicals (OH) (Ding et al., 2022). Plants have

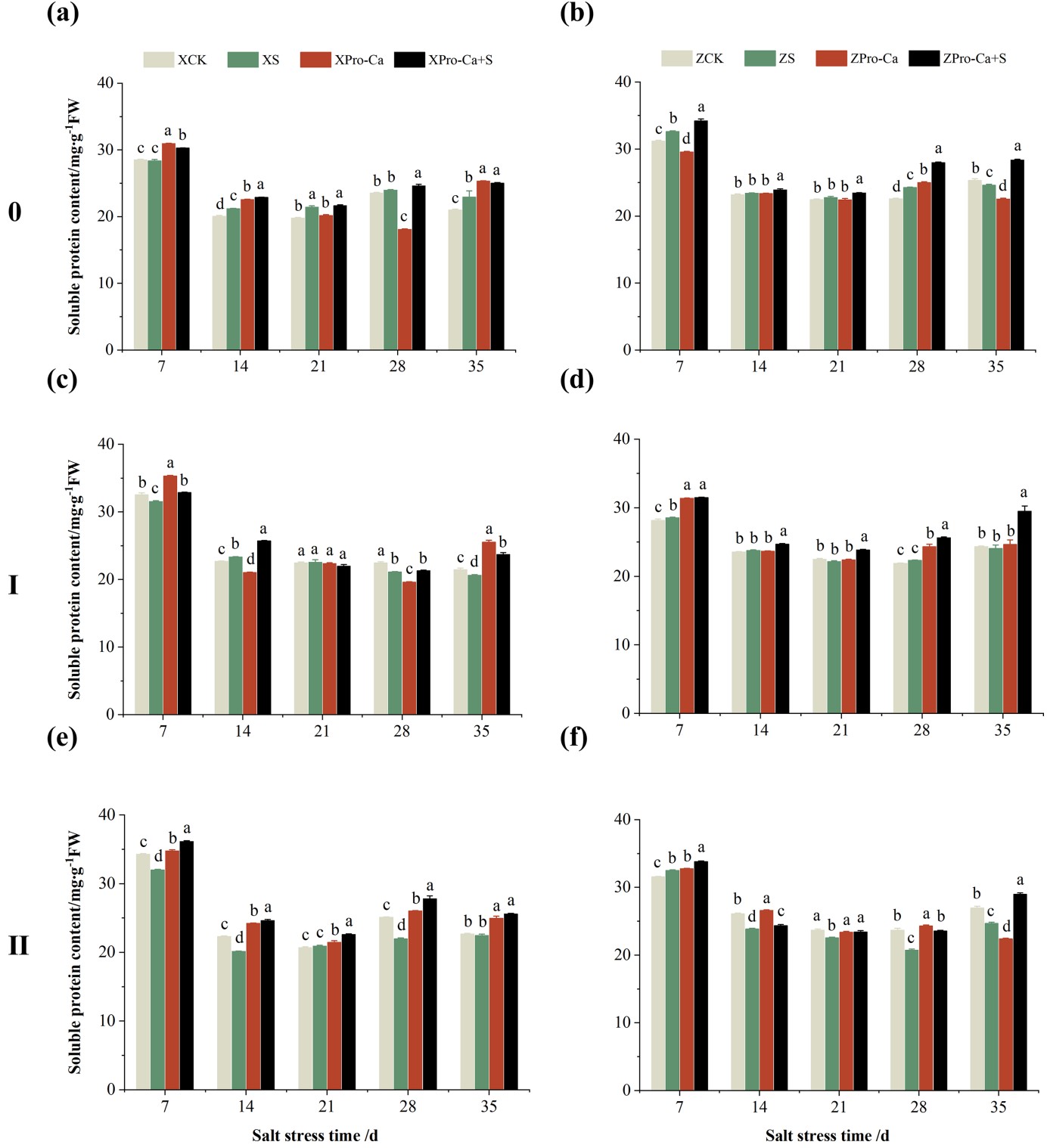

**Figure 16 Effect of Pro-Ca on the soluble protein content of rice main stem (A, B), first tiller (C, D), and second tiller (E, F) stems under salt stress.** The different letters are significant differences according to Duncan's new multiple range test ($p < 0.05$) based on one-way ANOVA. Xiangliangyou900: XCK (distilled water), XS (0.3% NaCl), XPro-Ca (100 mg·L$^{-1}$ Pro-Ca), XPro-Ca+S (100 mg·L$^{-1}$ Pro-Ca + 0.3% NaCl), Huanghuazhan: ZCK (distilled water), ZS (0.3% NaCl), ZPro-Ca (100 mg·L$^{-1}$ Pro-Ca), ZPro-Ca+S (100 mg·L$^{-1}$ Pro-Ca + 0.3% NaCl).

evolved complex mechanisms to combat salt stress-induced oxidative stress, such as antioxidant enzymes like SOD, CAT, and POD. These enzymes are crucial for scavenging ROS, and an elevation in ROS levels triggers an upsurge in the activities of SOD, POD, CAT, and other enzymes during salt stress conditions (*Huihui et al., 2020*). Salt stress damage to crops has been confirmed in various crops (*Kousar et al., 2021*; *Jung, Hütsch & Schubert, 2017*; *Feng et al., 2023*). *Jian et al. (2022)* demonstrated that growth rate was reduced after 1 day of 100 mM NaCl stress in rice, and more severe wilting symptoms appeared on the tips of rice plant leaves after 15 days. 'IR29' was particularly damaged, which contained high levels of ROS. The results of this experiment indicated that the main stem leaves exhibited a stronger antioxidant capacity under salt stress due to increased SOD, POD, and APX activities. CAT activity decreased in all tillers under salt stress, while the main stem demonstrated higher salt tolerance compared to the tillers by enhancing SOD, CAT, and POD enzyme activities and experiencing a less significant reduction in APX. It was speculated that this discrepancy could be attributed to a lower level of stress in the main stem and a greater resilience in the main stem in contrast to the tillers. *Yang et al. (2022)* suggested that since tillers mature later than the main stem, the main stem consistently maintains a growth and development advantage. Prolonged exposure to stress further reinforces this superiority, leading to unequal competition for C and N resources between the main stem and tillers, ultimately diminishing seed yield (*Tilley, Heiniger & Crozier, 2019*). *Xie et al. (2019)* observed that salt-tolerant wheat under salt stress conditions had higher activities of CAT, POD and APX thus more unfavourable $H_2O_2$ accumulation and reduced oxidative damage compared to sensitive varieties. Therefore, it is believed that the leaf tiller co-extension law leads to a pre-eminent advantage of temperature and light resource utilisation in the main stem and the first and second tillers, which contributes to the improvement of antioxidant capacity, nutrient accumulation and thus better growth and resistance.

Foliar spraying of Pro-Ca under salt stress effectively mitigated the damage of salt stress on morphogenesis at the tillering stage of rice, as demonstrated in our previous studies (*Zhang et al., 2023b*, *2023c*; *Huang et al., 2023*). In line with previous experiments, the results of this study indicated that foliar application of Pro-Ca had a more pronounced impact on tiller development compared to the main stem. Pro-Ca effectively alleviated the oxidative damage caused by salt stress on rice leaves. Foliar spraying of Pro-Ca increased the SOD and APX activities of tiller leaves and stems to a stronger extent than that of the main stems, but the effect on CAT activity of main stem leaves was more pronounced; moreover, Pro-Ca showed a better modulation in the alleviation of membrane damage and in the increase of soluble protein content. On the one hand, compared with the main stem, tillers will be subject to stronger oxidative damage, calcium regulator can increase SOD and APX activities as the first line of defence of the plant against oxidative stress (*Shafi et al., 2015*), on the other hand, it may be related to the spraying site of Pro-Ca, the treatment of the present experiment was in the seedling stage, and the sampling period was in the tillering stage, and the determined tillers were developed from the axils of the leaves that had been sprayed with Pro-Ca in the previous period, therefore, the regulator may have a better regulatory effect on the site to which it was sprayed. In prior trials, it was

discovered that despite not being directly treated with ethylene and chlormequat chloride, oat T1 and T2 tillers exhibited delayed stem growth. This could be attributed to the transition from chlormequat chloride-treated to untreated sections of the plant, a phenomenon also observed in wheat (*Kang et al., 2010*; *Peltonen-Sainio et al., 2003*).

## CONCLUSION

This experiment showed that salt stress hindered the growth of leaves and stems of rice tillers during the tillering stage. Additionally, the application of Pro-Ca through spraying effectively reduced the oxidative damage caused by salt stress on the tillers. Interestingly, the impact on the tillers was more pronounced compared to the main stems under similar conditions. This study offers new perspectives on the varying effects of salt stress on rice tillers and the regulatory role of Pro-Ca.

## ACKNOWLEDGEMENTS

We are very grateful to all authors for their contributions to this article. We would like to thank the editor and reviewers for their positive comments.

### Funding

This work was supported by the Special Project for Key Areas of General Colleges and Universities of Guangdong Provincial Department of Education (2021ZDZX4027), the Innovative Team Project of General Colleges and Universities of Guangdong Province (2021KCXTD011), the Zhanjiang Municipal Bureau of Science and Technology (2022A01016), and the Zhanjiang Innovation and Entrepreneurship Team Leadership Program (2020LHJH01). The funders had no role in study design, data collection and analysis, decision to publish, or preparation of the manuscript.

### Grant Disclosures

The following grant information was disclosed by the authors:
Special Project for Key Areas of General Colleges and Universities of Guangdong Provincial Department of Education: 2021ZDZX4027.
Innovative Team Project of General Colleges and Universities of Guangdong Province: 2021KCXTD011.
Zhanjiang Municipal Bureau of Science and Technology: 2022A01016.
Zhanjiang Innovation and Entrepreneurship Team Leadership Program: 2020LHJH01.

### Competing Interests

The authors declare that they have no competing interests.

### Author Contributions

- Rongjun Zhang conceived and designed the experiments, performed the experiments, analyzed the data, prepared figures and/or tables, and approved the final draft.

- Dianfeng Zheng conceived and designed the experiments, authored or reviewed drafts of the article, and approved the final draft.
- Naijie Feng conceived and designed the experiments, authored or reviewed drafts of the article, and approved the final draft.
- Linfeng Linfeng performed the experiments, prepared figures and/or tables, and approved the final draft.
- Jinning Ma performed the experiments, prepared figures and/or tables, and approved the final draft.
- Xiayi Yuan performed the experiments, prepared figures and/or tables, and approved the final draft.
- Junyu Huang performed the experiments, prepared figures and/or tables, and approved the final draft.
- Lisha Huang performed the experiments, prepared figures and/or tables, and approved the final draft.

## Data Availability

The raw measurements are available in the Supplemental File.

## Supplemental Information

Supplemental information for this article can be found online at http://dx.doi.org/10.7717/peerj.18357#supplemental-information.

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
