# Peer review of "Effect of salt stress on different tiller positions in rice and the regulatory effect of prohexadione calcium"

_PeerJ, doi:10.7717/peerj.18357_

## Round 0.1 · original submission · Major Revisions

The salinity of agricultural soil will be one of the most important issues that scientists may spend so much time on in the future. Therefore, your article has valuable data for the next studies. However, some parts of your article need to improve. I listed them below.

The methodology should be detailed to make it reproducible. Please add the details such as concentrations, stages, and calibration curves.

Please check your references and the reference list.

Each factor was evaluated in itself. Why did not you evaluate the interactions between time, treatment, and tillering stage? If you made it, you should have explained in the article.

Please carefully read the reviewers' suggestions about your article. If you do not accept one or more of their suggestions, give your reasons.

·

Basic reporting

No comment

Experimental design

# The author mentioned that four replicates were performed per treatment (line 168 to 174). However, n=3 was presented in the figure legend. Why not consistent?

# In each replicate, how many plants was measured? The information should be provided.

# All the methods should be described instead of only citation (line 224 to 226).

# Most physiology experiments were performed on 0.5g tissue. How many plants the tissue came from?

Validity of the findings

# The authors used two kinds of rice: Xiangliangyou900 and Huanghuazhan. What’s their difference? Have the author compared whether they had consistent morphology and physiology results?

# The authors had a conclusion “the main stem stem showed greater salt tolerance than the tillers under salt stress by increasing SOD, CAT, and POD enzyme activities” (line 602). However, when compare Figure 6a and 6c, the SOD enzyme activities in stem and tiller are similar on the 7d, 21d and 28d under salt stress. On the 14d and 35d, the tiller has more SOD activities. How the authors explain this?

Additional comments

In this manuscript, authors performed the morphology and physiology study of rice under salt stress and found that Pro-Ca alleviated the effects caused by salt stress. The authors conducted a large amount of data collection and analysis, although there is no mechanism study at all. The conclusions are for the most part supported by the data. However, there are several concerns that should be addressed prior to publication.

·

Basic reporting

I think this manuscript brings a certain novelty to the area.
Some corrections and suggestions were made, always aiming to improve the study.

All observations are highlighted in yellow in the text.

Experimental design

Some points need to be detailed.

Validity of the findings

The present study brings new analyzes in the area.

Additional comments

The English language must be improved.

---

## Round 0.2 · Minor Revisions

I appreciate your positive and constructive attitude toward the reviewers' suggestions. However, your article requires some minor revisions before publication. Please carefully read the reviewers' comments and consider each of them. If you disagree with any particular suggestion, it would be beneficial to provide clear and well-reasoned justifications for your perspective. Additionally, could you please add the raw data from your study as a supplemental file?

·

Basic reporting

Nothing to declare.

Experimental design

Nothing to declare.

Validity of the findings

Nothing to declare.

Additional comments

Nothing to declare.

---

## Round 0.3 · accepted · Accept

I appreciate your positive and constructive attitude toward the suggestions of reviewers. I believe your manuscript is now ready for publication. We look forward to your next article.